# Reorganization of postmitotic neuronal chromatin accessibility for maturation of serotonergic identity

Xinrui L Zhang[†], William C Spencer[†], Nobuko Tabuchi, Meagan M Kitt, Evan S Deneris*

Department of Neurosciences, Case Western Reserve University, Cleveland, United States

**Abstract** Assembly of transcriptomes encoding unique neuronal identities requires selective accessibility of transcription factors to cis-regulatory sequences in nucleosome-embedded postmitotic chromatin. Yet, the mechanisms controlling postmitotic neuronal chromatin accessibility are poorly understood. Here, we show that unique distal enhancers define the *Pet1* neuron lineage that generates serotonin (5-HT) neurons in mice. Heterogeneous single-cell chromatin landscapes are established early in postmitotic *Pet1* neurons and reveal the putative regulatory programs driving *Pet1* neuron subtype identities. Distal enhancer accessibility is highly dynamic as *Pet1* neurons mature, suggesting the existence of regulatory factors that reorganize postmitotic neuronal chromatin. We find that Pet1 and Lmx1b control chromatin accessibility to select *Pet1*-lineage-specific enhancers for 5-HT neurotransmission. Additionally, these factors are required to maintain chromatin accessibility during early maturation suggesting that postmitotic neuronal open chromatin is unstable and requires continuous regulatory input. Together, our findings reveal postmitotic transcription factors that reorganize accessible chromatin for neuron specialization.

*For correspondence:
esd@case.edu

[†]These authors contributed equally to this work

Competing interest: The authors declare that no competing interests exist.

## Editor's evaluation

This study will be of interest to developmental biologists who study the gene regulatory mechanisms necessary for induction and maintenance of post-mitotic neuronal identity. The authors generated a useful resource of genomic data and provided new insights into the dynamic regulation of accessible chromatin regions in post-mitotic serotonin (5-HT) neurons of the mouse hindbrain. This work proposes two transcription factors (Pet1, Lmx1b) as necessary for establishment and maintenance of accessible chromatin regions in serotonin neurons.

## Introduction

Specialized neuronal identities are acquired through the differential expression of genes that specify transmitter usage, build dendritic and axonal architecture, and control synapse formation (*Hobert, 2011*). The maturation of neuronal functions is an extended process that plays out over several weeks of mid fetal to early postnatal life in mice and through the first few years of life in humans (*Stiles and Jernigan, 2010*), as postmitotic neuronal transcriptomes mature to a stable state. Unique combinations of sequence-specific transcription factors (TFs) selectively activate and repress genes to assemble the transcriptomes that encode individual neuron identities (*Polleux et al., 2007*). Genes with variants associated with autism and other neurodevelopmental disorders exhibit enriched expression in maturing neurons and are often involved in chromatin remodeling and transcriptional regulation; their

developmental dysfunction is thought to contribute to disease pathogenesis (*De Rubeis et al., 2014*; *Heavner and Smith, 2020*).

Selective expression of neuronal identity genes depends on selective accessibility of promoter and enhancer sequences embedded in postmitotic nucleosomal-organized chromatin to regulatory factor inputs (*Gallegos et al., 2018*; *Perino and Veenstra, 2016*). Chromatin accessibility is dynamic as brain regions (*Gorkin et al., 2020*; *de la Torre-Ubieta et al., 2018*) and neurons (*Frank et al., 2015*; *Preissl et al., 2018*; *Stroud et al., 2020*; *Trevino et al., 2021*) mature. Yet, we have a poor understanding of how the dynamics of accessible chromatin in postmitotic neurons prefigures acquisition of neurotransmitter identities and myriad other molecular and morphological features of specialized neurons. Further, the regulatory factors involved in selecting postmitotic accessible chromatin regions to enable sequence-specific activation of unique combinations of identity features have not been identified.

Control of 5-HT neuron development is of particular interest as 5-HT has expansive modulatory effects on central neural circuitry (*Celada et al., 2013*). Altered serotonergic gene expression, brought about by either genetic or environmental factors, has been implicated in neuropsychiatric disorders including depression, stress-related anxiety disorders, autism, obsessive compulsive disorder, and schizophrenia (*Deneris and Wyler, 2012*). Postmitotic 5-HT neuron precursors terminally differentiate in the ventral hindbrain to acquire the capacity for 5-HT synthesis, reuptake, vesicular transport, and degradation through the coordinate expression of the 5-HT neurotransmission genes *Tph2*, *Ddc*, *Gch1*, *Slc6a4* and *Slc22a3*, *Slc18a2*, and *Maoa/b* (*Deneris and Gaspar, 2018*). Subsequently, newborn 5-HT neurons enter a maturation stage during which serotonergic transcriptomes are highly dynamic as 5-HT neurons refine their terminal identities, migrate to populate the various raphe nuclei, acquire mature firing properties, and establish expansive connectivity throughout the brain and spinal cord (*Deneris and Gaspar, 2018*).

The ETS domain factor Pet1 and the LIM homeodomain (HD) factor Lmx1b are terminal selector TFs in 5-HT neurons as they are continuously expressed beginning at the postmitotic precursor stage, control their own expression through direct positive autoregulation, and are required for the acquisition of 5-HT neuron terminal identity through sequence-specific transcriptional activation of 5-HT pathway genes (*Deneris and Gaspar, 2018*; *Hobert, 2008*). Pet1 and Lmx1b are also required for the acquisition of serotonergic firing characteristics and formation of long-distance profusely arborized 5-HT axon architectures throughout the brain and spinal cord (*Donovan et al., 2019*; *Liu et al., 2010*; *Wyler et al., 2016*). Pet1's function in 5-HT neurons is clinically significant as recent studies identified biallelic loss of function of *FEV*, the human ortholog of *Pet1*, in autism cases suggesting that defects in transcriptional control of 5-HT neurons are a potential path to human neurodevelopmental disorders (*Doan et al., 2019*).

The low abundance of many postmitotic neuron types has hindered the assay of their accessible chromatin landscapes and cis-regulatory architectures. Here, we adapted bulk and single-cell ATAC-seq (scATAC-seq), scRNA-seq, and ChIPmentation protocols to map accessible chromatin and cis-regulatory elements (CREs) of developing 5-HT neurons. We then used these combined maps along with Pet1 and Lmx1b knockout mouse models and occupancy mapping to identify a previously unrecognized function of these TFs in regulating the chromatin accessibility of 5-HT neurotransmission and synapse genes. Together, our findings show that Pet1 and Lmx1b control the maturation of 5-HT connectivity not only through sequence-specific activation of select terminal effector genes but also by reorganizing postmitotic accessible chromatin at putative CREs.

## Results

### The unique accessible chromatin landscape of maturing postmitotic *Pet1* neurons

5-HT neurons are generated from *Pet1*-positive postmitotic precursors located in the embryonic ventral hindbrain from the isthmus to the caudal region of the myelencephalon (*Alonso et al., 2013*; *Hendricks et al., 1999*). Therefore, we used the *Pet1-EYFP* transgenic mouse line to flow sort E14.5 Yfp-positive neurons (*Pet1* neurons) from the entire longitudinal extent of the hindbrain for assay of accessible chromatin in maturing 5-HT neurons (*Scott et al., 2005a*; *Scott et al., 2005b*; *Figure 1— figure supplement 1a-c*). We generated ATAC-seq biological replicates with a mean of 29M unique

paired-end mapped reads per sample (*Figure 1—figure supplement 1d-g*) and identified 68,871 peaks of transposase accessible chromatin (TAC) at E14.5, and 59,323 TACs in an equal number of control Yfp-negative, *Pet1*-negative cells ('*Pet1*^neg') collected from the same hindbrain regions at the same developmental age, which comprise diverse non-serotonergic cell types. Principal component analysis (PCA) showed that cell type identity accounted for 93% of the variance when comparing *Pet1* neuron ATAC-seq replicates with *Pet1*^neg ATAC-seq samples and with previously published ATAC-seq datasets obtained from bulk tissue dissections (*Figure 1a*; *Gorkin et al., 2020*). The accessible chromatin landscape of *Pet1* neurons is distinct from previously published bulk hindbrain ATAC-seq (*Gorkin et al., 2020*), which indicates that tissue level analyses did not capture the chromatin accessibility profiles of low abundance *Pet1* neurons (*Figure 1a*). Indeed, of the TACs we identified in *Pet1* neurons at E14.5, 31% are unique (*Pet1* neuron-specific TACs) and not found in *Pet1*^neg cells and 22% are not previously identified in any other cell type (*Gorkin et al., 2020*; *Figure 1b*).

To determine the functions of 5-HT neuron TACs, we next used ChIPmentation (*Schmidl et al., 2015*) to profile the genome-wide distribution of histone posttranslational modifications associated with transcriptional activation or repression in flow sorted E14.5 *Pet1* neurons. With the segmentation algorithm ChromHMM (*Ernst and Kellis, 2017*), we defined 10 chromatin states by assessing the combinatorial pattern of histone marks at each genomic locus (*Figure 1c*; *Figure 1—figure supplement 1h-i*). Active and poised distal enhancer regions are over-represented among the *Pet1* neuron-specific TACs (*Figure 1c*). Our ChIPmentation analysis also identified 1919 genomic regions that fulfill the criteria for super-enhancers based on the presence of unusually expansive H3K27ac signals (*Figure 1d*; *Lovén et al., 2013*; *Whyte et al., 2013*). Comparing these to previously annotated enhancers in other cell types (*Gorkin et al., 2020*), we found that a significant percentage (42%) of the super-enhancers we detected in *Pet1* neurons were not detected in bulk mouse forebrain, hindbrain, or liver tissues (*Figure 1e*), suggesting that they are highly specific to the *Pet1* lineage. As an example, a 82.9 kb super-enhancer detected in *Pet1* neurons but not *Pet1*^neg neurons spans the *Lmx1b* locus (*Figure 1f*).

Notably, a distinct TAC was found immediately upstream of the *Pet1* promoter in *Pet1* neurons but not in *Pet1*^neg cells (*Figure 1g*). This TAC shows enrichment of H3K4me1 and H3K27ac marks suggesting the presence of an active enhancer (*Figure 1g*). Previous transgenic assays, in vivo, demonstrated that this upstream region directs robust reporter expression to developing and adult 5-HT neurons present in each of the midbrain, pontine, and medullary raphe nuclei (*Krueger and Deneris, 2008*; *Scott et al., 2005a*; *Scott et al., 2005b*), hence providing functional validation of serotonergic enhancers we have defined with ATAC-seq and ChIPmentation. As further validation, we intersected all identified *Pet1* neuron TACs with the VISTA enhancer database (*Visel et al., 2007*). Of all experimentally validated VISTA enhancers that show activity in the hindbrain, 38% overlap with *Pet1* neuron TACs (*Figure 1—figure supplement 1j*). However, only 1% of hindbrain-annotated VISTA enhancers overlap with *Pet1* neuron-specific TACs, thus demonstrating that analyses of chromatin accessibility in low abundance neuron types can reveal previously unrecognized CREs (*Figure 1—figure supplement 1j*). Importantly, genes associated with *Pet1* neuron-specific TACs are enriched for axon and synapse functions (*Figure 1h*). Indeed, we found *Pet1* neuron-specific TACs for 5-HT neurotransmission genes (*Figure 1i–j*). Together, our ATAC-seq and ChIPmentation analyses define the open chromatin landscape of maturing 5-HT neurons and reveal that distal TACs putatively assigned to genes encoding axon and synapse-related functions define the *Pet1* postmitotic neuron lineage.

## Heterogeneous TAC accessibility and gene expression define *Pet1* neuron subtypes at embryonic stages

Adult 5-HT neurons possess diverse transcriptomes that encode functional subtypes (*Okaty et al., 2015*; *Okaty et al., 2020*; *Ren et al., 2019*; *Wylie et al., 2010*), yet the development of 5-HT neuron heterogeneity is poorly understood. To determine whether the 5-HT neuron accessible chromatin is heterogeneous early in development, we carried out scATAC-seq using E14.5 flow sorted *Pet1* neurons, which were processed using the 10× Genomics Chromium microfluidic droplet method. The resulting scATAC-seq library was sequenced to a depth sufficient to yield >50,000 paired-end reads per nucleus. After filtering based on standard quality control measures (*Figure 2—figure supplement 1a*), we retained 1692 nuclei for analysis. This led to the identification of 124,111 TACs, 60,117 of which were not identified in our bulk *Pet1* neuron ATAC-seq (*Figure 2a*). We observed high

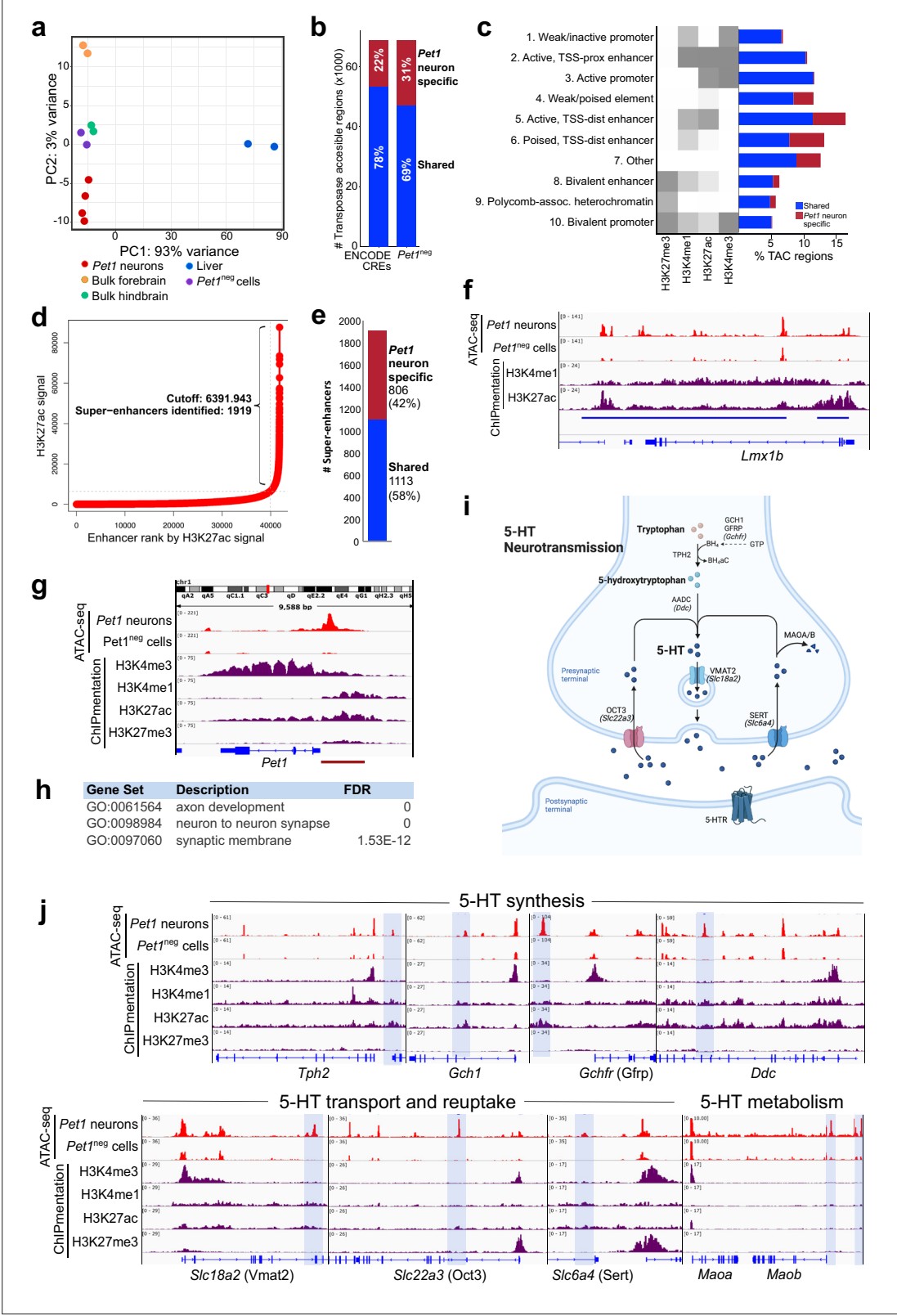

**Figure 1.** Unique distal enhancers and super-enhancers define the *Pet1* lineage. (**a**) Principal component analysis (PCA) reveals the unique accessible chromatin landscape of *Pet1* neurons. Dots of the same color represent biological replicates. (**b**) Bar plot showing percentage of unique ATAC-seq peaks (transposase accessible chromatin [TAC] regions) from E14.5 *Pet1* neurons when compared to cis-regulatory elements (CREs) annotated by ENCODE datasets (*Gorkin et al., 2020*) (left bar) or with transposase accessible chromatins (TACs) in E14.5 Pet1-negative (*Pet1neg*) cells from the same

*Figure 1 continued on next page*

*Figure 1 continued*

hindbrain region (right bar). (**c**) Emission probabilities for histone modifications in 10 ChromHMM states and the percent genomic coverage of each chromatin state for E14.5 *Pet1* neuron-specific TACs and *Pet1* neuron TACs shared with other tissues in ENCODE ATAC-seq datasets.(**d**) Distribution of H3K27ac signal across all enhancers identified in E14.5 *Pet1* neurons. Enhancers with exceptionally high read counts (bracket) were designated super-enhancers. (**e**) Bar plot of *Pet1* neuron super-enhancers that are unique to *Pet1* neurons or overlap ('shared') with annotated enhancers from ENCODE datasets. (**f**) Genome browser view of the *Pet1* neuron-specific super-enhancer across the *Lmx1b* locus defined as an extended stretch of H3K27ac and H3K4me1 enrichment.(**g**) Genome browser views of ATAC-seq and ChIPmentation signals at the *Pet1/Fev* locus. Red bar denotes the conserved *Pet1* enhancer region that is sufficient to direct 5-HT neuron-specific transgene expression (*Scott et al., 2005a*; *Scott et al., 2005b*; *Krueger and Deneris, 2008*). (**h**) Gene ontology (GO) of *Pet1* neuron-specific TACs (not found in other tissues in ENCODE ATAC-seq datasets). (**i**) Diagram of 5-HT neurotransmission. (**j**) Genome browser views of ATAC-seq and ChIPmentation peaks at 5-HT neurotransmission genes. Light blue shading highlights *Pet1* neuron-specific TACs.

The online version of this article includes the following figure supplement(s) for figure 1:

**Figure supplement 1.** ATAC-seq quality control and analysis.

concordance (Spearman's correlation = 0.9) between our pseudo-bulk scATAC-seq dataset and our bulk ATAC-seq data as illustrated by the highly similar TAC patterns in the *Pet1* upstream enhancer region (*Figure 2b*). In parallel we also performed scRNA-seq using E14.5 flow sorted *Pet1* neurons. We obtained over 80,000 reads per cell for 1612 cells, with a median of 4010 genes per cell and a total of 19,707 genes across all cells (*Figure 2—figure supplement 1b*); 95% of the flow sorted *Pet1* neurons robustly expressed both *Pet1* and *Yfp*, validating that our cell sorting method yields nearly pure samples of *Pet1* neurons. The small numbers of *Yfp*-negative cells were removed from further analysis.

We grouped single *Pet1* neurons to identify 11 major subtypes with unique chromatin accessibility and expression patterns (*Figure 2d and f*; *Figure 2—figure supplement 1c-d*). *Pet1* and *Lmx1b* loci TACs are robustly accessible in virtually all cells within the 11 identified subtypes (*Figure 2e*). Like *Pet1* and *Lmx1b*, two-thirds (63%) of TACs are broadly accessible among multiple subtypes, including TACs of 5-HT neurotransmission genes *Tph2*, *Slc22a3*, and *Slc6a4* (*Figure 2e*). In contrast, one-third of TACs (36.6%) are unique to a single cluster. Subtype-specific TACs are predominantly novel distal gene enhancers as previously defined by chromatin state (*Figure 1c*; *Figure 2—figure supplement 1e-g*) and are often observed near genes that mark known 5-HT neuron subtypes (*Figure 2c*; *Figure 2—figure supplement 1c*). For example, TF En1 is enriched in the rostral rhombencephalic 5-HT neurons, whereas Hox gene cofactor Meis2 is selectively expressed in the caudal rhombencephalic 5-HT neurons (*Wylie et al., 2010*). Six of the scATAC-seq clusters share a unique TAC region immediately upstream of the TSS of the rostral marker gene *En1* while the remaining five clusters have more accessible TAC regions near the promoter of the caudal 5-HT neuron marker gene *Meis2* (*Figure 2c–e*). To compare our embryonic scRNA-seq clustering results to published adult scRNA-seq data, we performed a unimodal uniform manifold approximation and projection (UMAP) of the adult 5-HT scRNA-seq data from *Okaty et al., 2020*, onto our E14.5 dataset (*Figure 2—figure supplement 1h*). Since the adult scRNA-seq data was obtained from the dorsal raphe nuclei (DRN), which is solely derived from the isthmus and rhombomere 1, we predicted that the adult scRNA-seq data would largely map to our rostral clusters expressing high levels of *Tph2* and *En1*. Indeed, 93% of the adult DRN cells map to the E14.5 *Tph2/Slc6a4* and *Pax5/En1* clusters that have high expression of rhombomere 1 markers (*Figure 2—figure supplement 1h*). Thus, our data demonstrate that diverse single-cell 5-HT neuron chromatin landscapes and transcriptomic profiles are established early in postmitotic *Pet1* neurons and likely prefigure the heterogeneity of adult 5-HT neurons at least in the DRN.

## Mapping accessible chromatin regions to genes

To analyze the association between CRE accessibility and target gene expression, we co-embedded scATAC-seq with scRNA-seq data, then linked TACs to a gene if logistic regression detected a statistical association between gene expression and the binarized chromatin accessibility at a genomic region within 500 kb of a gene (*Fang et al., 2021*). Applying this method to our single-cell datasets yielded a total of 41,191 TAC-gene pairings (*Source data 1*). At this developmental time point, each gene is on average linked to only 1.8 TACs and most genes are associated with either none or a single TAC (*Figure 2h*). However, 10% of genes are associated with >5 TAC regions, and a select group of genes (0.045%) are associated with >20 TACs (*Figure 2h–j*; *Figure 2—figure supplement 2a*). Gene

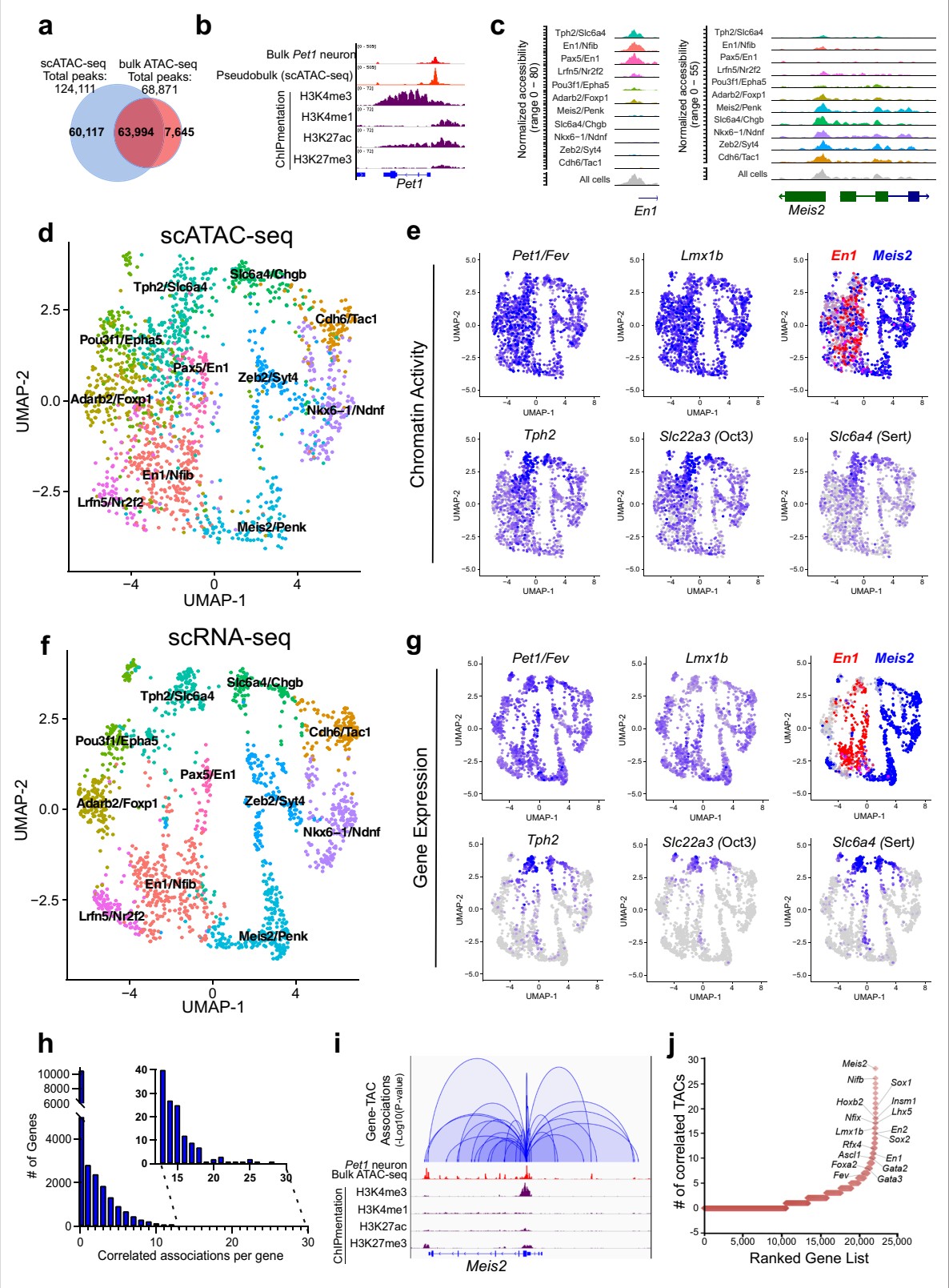

**Figure 2.** Heterogeneity in chromatin accessibility and transcriptomes reveals transcriptional programs determining *Pet1* neuron subtype identities. (**a**) Venn diagram showing overlap of single-cell *Pet1* neuron TACs vs. bulk ATAC-seq TACs. (**b**) Genome browser view of *Pet1* locus showing high concordance between the pseudobulk scATAC-seq and bulk FACS-purified *Pet1* neuron ATAC-seq at E14.5. (**c**) scATAC-seq tracks showing the aggregated chromatin accessibility peaks of *En1* (left) and *Meis2* (right) for each cluster. (**d**) Uniform manifold approximation and projection (UMAP)

*Figure 2 continued on next page*

*Figure 2 continued*

visualization of chromatin accessibility of single E14.5 *Pet1*-lineage neurons. scATAC-seq is projected onto scRNA-seq UMAP space. Cells are colored by scATAC-seq cluster assignment. Genes associated with the top two marker TACs for each cluster are labeled. (**e**) The chromatin accessibility profiles of select genes across UMAP visualized scATAC-seq clusters. (**f**) UMAP visualization of single E14.5 *Pet1* neurons based on gene expression. Cells are colored by scRNA-seq cluster assignment. (**g**) The expression profile of example 5-HT neuron identity genes across all scRNA-seq clusters in UMAP. (**h**) Number of TAC-gene associations for all genes. (**i**) Representative gene with high number of associated TACs. Loops represent statistically significant TAC-gene association. Loop height represents the p-value of TAC-gene correlation. (**j**) Number of significantly correlated TACs for each gene. Genes are ranked by the numbers of associated peaks. Select TFs with known or potential functions in the *Pet1* lineage are labeled.

The online version of this article includes the following figure supplement(s) for figure 2:

**Figure supplement 1.** Identification of *Pet1* neuron subtype-specific TACs.

**Figure supplement 2.** *Pet1* neuron subtype-specific TF activity, expression, and accessibility.

ontology (GO) analysis of the top 1500 genes with the highest number of linked CREs revealed that they are highly enriched for genes involved in axon and synapse development (***Figure 2—figure supplement 2b***), suggesting that a large number of CREs converge to fine-tune or safeguard the transcription of key neuronal regulators (***Ma et al., 2020***; ***Trevino et al., 2021***). Among these genes are known TFs of the 5-HT neuron gene regulatory network such as Pet1 and Lmx1b (***Figure 2j***).

We identified many subtype-specific TFs constituting a potential TF code that directs 5-HT neuron heterogeneity (***Figure 2—figure supplement 2c-d***). As examples, *Nkx6-1* gene expression, chromatin accessibility, and ChromVAR-inferred binding at NK motifs are enriched in the Nkx6-1/Ndnf cluster, while expression, chromatin accessibility, and inferred binding of the homeodomain TF Meis2 are enriched in the Meis2/Penk cluster (***Figure 2—figure supplement 2c-d***; ***Schep et al., 2017***). In contrast, the expression and accessibility of TFs such as Pet1/Fev and Lmx1b are more evenly distributed among all *Pet1* neuron subtypes (***Figure 2e and g***; ***Figure 2—figure supplement 2d***). Overall, most subtype-specific TACs are correlated with the subtype-specific expression pattern of their linked genes; however, we noticed that the expression of 5-HT identity genes *Tph2*, *Slc6a4*, and *Slc22a3* are low or undetectable in most *Pet1* neurons at E14.5 despite their relatively uniform accessible chromatin in *Pet1* neurons at this stage (***Figure 2e and g***). The vast majority of *Pet1*-lineage neurons robustly express these 5-HT identity genes in the adult mouse brain (***Figure 2—figure supplement 2e***; ***Donovan et al., 2019***; ***Okaty et al., 2015***; ***Okaty et al., 2019***; ***Ren et al., 2019***). Tph2 immunofluorescence analysis at E14.5 revealed that Tph2-positive cells emerge across all raphe nuclei with particular enrichment in the dorsal raphe (***Figure 2—figure supplement 2f***). This suggests that chromatin accessibility may be remodeled early during maturation to prime neuron identity before the transcription of terminal effector genes.

## A highly dynamic stage of postmitotic accessible chromatin reorganization

We next investigated whether *Pet1* neurons are born with fully mature chromatin that is accessible to all regulatory factors needed for the eventual transcription of 5-HT genes. In this case, developmental gene expression trajectory is likely supported by the dynamic interactions of transcriptional regulatory factors with DNA binding sites. Alternatively, *Pet1* neurons are perhaps born with immature chromatin that must be remodeled to generate the mature transcriptomes needed to encode adult characteristics. We performed bulk *Pet1* neuron ATAC-seq (***Figure 3—figure supplement 1a-b***) to compare TACs at E11.5 when postmitotic 5-HT neuron precursors are predominant, E14.5 when *Pet1* neurons are actively extending dendrites and axons, and E17.5 when *Pet1* neurons are coalescing to form the raphe nuclei and innervating distal target brain regions (***Donovan et al., 2019***). PCA showed high concordance of ATAC-seq replicates within each time point with developmental state accounting for the majority of variance (***Figure 3a***). We found that significant gains and losses in chromatin accessibility of TSS-distal TACs occurred throughout embryonic maturation (***Figure 3b–d***; ***Figure 3—figure supplement 1c-d,i***). Of the TACs detected in *Pet1* neurons, 16,493 open, that is, gain accessibility (fold change >2 and FDR < 0.01) between E11.5 and E14.5, while 4148 open from E14.5 to E17.5 (***Figure 3c–d***). Concurrently, 9,402 E11.5 TACs close, that is, lose accessibility (fold change >2 and FDR < 0.01) by E14.5, and 1504 TACs close between E14.5 and E17.5 (***Figure 3c–d***). This suggests that chromatin remodeling is highly dynamic immediately following cell cycle exit and then gradually stabilizes. Assessment of chromatin states showed that active enhancers are over-represented

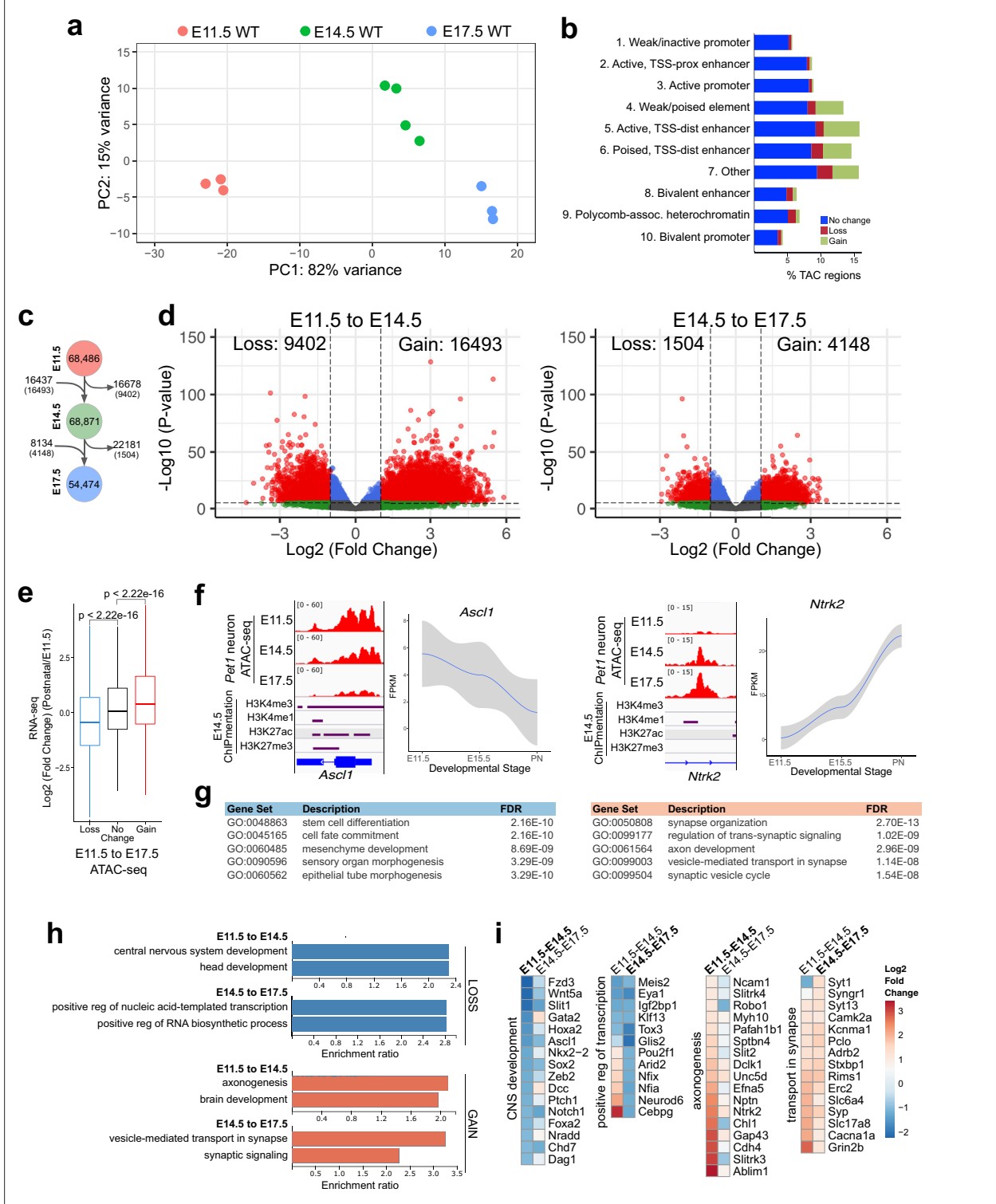

**Figure 3.** Gene regulatory dynamics in maturing *Pet1* neurons. (**a**) PCA showing the distribution of *Pet1* neuron ATAC-seq data over two principal components. (**b**) Distribution of chromatin states that exhibit gain (green), loss (red), or no change (blue) in chromatin accessibility from E11.5 to E17.5. (**c**) Number of TACs detected at each developmental time point (circles) and the number of TACs that are gained, lost, or propagated to the next time point (arrows). In parenthesis are the numbers of TACs that gain or lose >2-fold accessibility with FDR < 0.01. (**d**) Volcano plots showing the differential chromatin accessibility of TACs between consecutive time points, with fold change cutoff of 2-fold and significance cutoff of FDR < 0.01. Each dot represents one ATAC-seq peak (TAC). (**e**) Box plots showing the distribution of the log2 expression values of the linked genes for TACs that exhibit loss (blue), no change (black), or gain (red) in accessibility between E11.5 and E17.5. Significance of accessibility to gene expression correlation is calculated

*Figure 3 continued on next page*

Figure 3 continued

by Wilcoxon rank sum test. (**f**) Genome browser view of ATAC-seq and E14.5 ChIPmentation signals at representative genes that decrease (e.g. *Ascl1*) or increase (e.g. *Ntrk2*) in accessibility between E11.5 and E17.5. Parallel trends are seen in gene expression trajectories (panels on the right of tracks). (**g**) Top five GO biological process enrichment terms for genes linked to TACs that lose (left) or gain (right) accessibility from E11.5 to E17.5. (**h**) Bar graphs showing the enrichment ratio of the top gene ontology biological process enrichment terms for genes linked to TACs that lose (blue) or gain (orange) accessibility from E11.5 to E14.5 and from E14.5 to E17.5. (**i**) Heatmap showing the log2 fold change in chromatin accessibility for TACs of select genes associated with the indicated GO biological process terms for each developmental stage. Representative genes in GO terms from (**h**) were selected for display in (**i**).

The online version of this article includes the following figure supplement(s) for figure 3:

**Figure supplement 1.** Dynamic developmental remodeling of distal cis-regulatory elements.

among TACs opening between E11.5 and E17.5 (*Figure 3b*; *Figure 3—figure supplement 1e-g*). *Pet1* neuron super-enhancer accessibility is particularly highly dynamic during postmitotic neuron maturation, with the vast majority (93%) of TACs either gaining or losing accessibility between E11.5 and E17.5 (*Figure 3—figure supplement 1h*).

Next we compared our bulk ATAC-seq datasets at E11.5, E14.5, and E17.5 with our previously published bulk 5-HT neuron RNA-seq data obtained at E11.5, E15.5, and postnatal days 2–3 (*Wyler et al., 2016*). Using our TAC-to-gene assignments derived from single-cell data (*Figure 2i–j*), we found that changes in chromatin accessibility at cis-regulatory regions are significantly correlated with the temporal expression trajectories of their target genes (*Figure 3e–f*). GO analysis revealed that genes associated with closing TACs are important for neural precursor functions, whereas the genes linked to opening TACs are strongly enriched for axon development between E11.5 and E14.5 and synaptic signaling between E14.5 and E17.5 (*Figure 3g–i*). Dynamic TACs are highly enriched for diverse TF binding motifs, suggesting that combinatorial TF activities contribute to *Pet1* neuron maturation (*Figure 3—figure supplement 1j*). Together these findings support a model in which the dynamic remodeling of CRE accessibility drives gene expression trajectories required for maturation of 5-HT identity while shutting down expression of genes required at earlier stages of development.

## Pet1 reorganizes chromatin accessibility of 5-HT neurotransmission genes

The highly dynamic reorganization of TACs in maturing postmitotic *Pet1* neurons predicts the existence of regulatory factors that shape postmitotic neuronal chromatin landscapes. However, such factors have not been described. Notably, TACs that are unique to *Pet1*-lineage neurons are enriched for ETS-factor binding sites (*Figure 4—figure supplement 1a*). Thus, we hypothesized that Pet1 controls acquisition of 5-HT identity by controlling chromatin accessibility. *Pet1*⁻/⁻ 5-HT neurons are an ideal model with which to examine TF function in shaping the chromatin, as these neurons are retained in the brain in normal numbers and *Pet1-EYFP* transgene expression in these cells is robustly retained at fetal stages to permit their isolation and analysis (*Wyler et al., 2016*). PCA of ATAC-seq datasets obtained from *Pet1-EYFP*-positive, *Pet1*⁻/⁻ hindbrain cells at E11.5, E14.5, and E17.5 (*Figure 4—figure supplement 1b-c*) showed that Pet1-deficient *Pet1*-lineage neurons (*Pet1*⁻/⁻ neurons) cluster closely with wildtype *Pet1* neurons (*Figure 4a*; *Figure 4—figure supplement 1d*), consistent with our observation that Pet1-deficient 5-HT neurons do not adopt an overt alternative identity. However, we found a striking overall reduction of Pet1 footprints and decreased accessibility of many TSS-distal TACs in *Pet1*⁻/⁻ neurons compared to wildtype neurons at all three embryonic stages (*Figure 4b–c*; *Figure 4—figure supplement 1e*). To determine whether Pet1 directly controls chromatin accessibility in these chromatin regions, we performed CUT&RUN with E14.5 flow sorted *Pet1* neurons to generate a genomic occupancy map for Pet1 (*Figure 6—figure supplement 1a-c*). The CUT&RUN map significantly overlapped with our previous ChIP-seq of Pet1 (*Figure 4—figure supplement 1f*) but identified occupancy at several additional 5-HT neurotransmission genes that are controlled by Pet1 (*Figure 4i–j*; *Wyler et al., 2016*). CUT&RUN data showed that Pet1 directly occupies at least 53% of Pet1-dependent TACs (*Figure 4d–e*). Moreover, 73% of all Pet1-regulated TACs and 80% of Pet1-regulated TACs with Pet1 occupancy are TACs unique to *Pet1* neurons (*Figure 4f*).

Pet1-regulated TACs are predominantly those shared among multiple subtypes of *Pet1*-lineage neurons (*Figure 4h*). GO analysis revealed that the Pet1 occupied TACs that close in *Pet1*⁻/⁻ neurons are associated with genes required for serotonin synthesis (*Figure 4g*; *Figure 4—figure supplement*

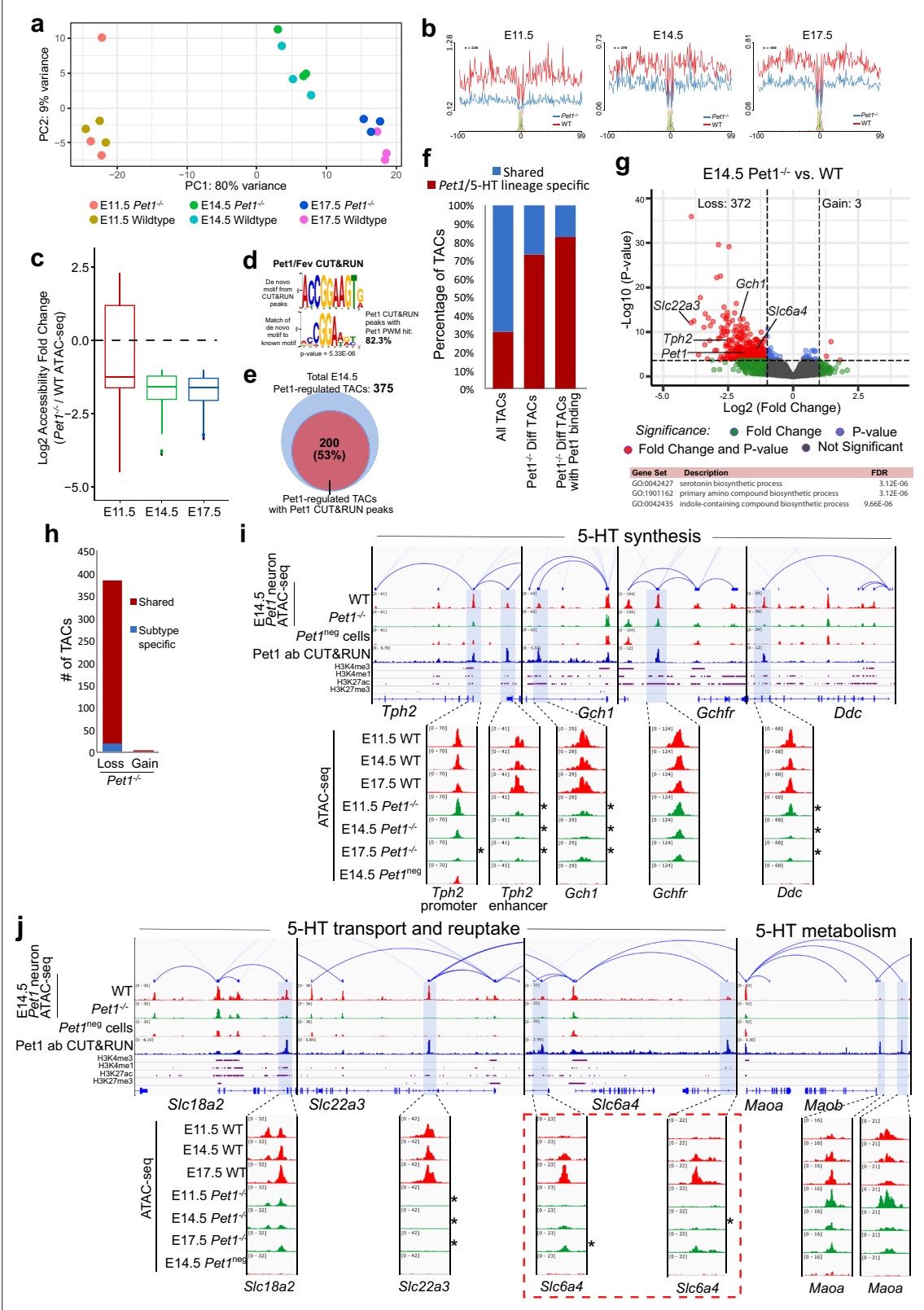

**Figure 4.** Pet1 reorganizes postmitotic chromatin accessibility. (**a**) PCA showing the distribution of ATAC-seq data for wildtype and *Pet1⁻/⁻ Pet1*-lineage neurons (*Pet1⁻/⁻* neurons) over two principal components. (**b**) In vivo Pet1 footprints derived from Tn5 insertion frequency (ATAC-seq reads) over representative ETS-factor motifs within *Pet1* neuron accessible chromatin regions. *Pet1⁻/⁻* neurons show reduced Pet1 footprints at E11.5 (left), E14.5 (middle), and E17.5 (right) compared to wildtype neurons for Pet1-regulated TACs. (**c**) Box plots showing the distribution of the log2 fold change in

*Figure 4 continued on next page*

*Figure 4 continued*

accessibility of the TACs that show differential accessibility between *Pet1⁻/⁻* vs. wildtype *Pet1* neurons at three developmental time points. *Pet1⁻/⁻* neurons show an overall decrease in chromatin accessibility. (**d**) The top significantly enriched motif within Pet1 CUT&RUN peaks as determined by de novo MEME motif analysis significantly matches (p = 5.33e-06) the Pet1/FEV high-affinity binding site previously defined in vitro (***Wei et al., 2010***) based on analysis by the TOMTOM Motif Comparison Tool (top panel). (**e**) The fraction of Pet1-regulated TACs containing Pet1 CUT&RUN peaks. (**f**) The fraction of all *Pet1* neuron TACs, *Pet1* neuron TACs that are dependent on Pet1, and Pet1-dependent TACs occupied by Pet1 that are specific to *Pet1*/5-HT lineage or shared with *Pet1ⁿᵉᵍ* cells. A significant fraction of Pet1-regulated TACs is unique to *Pet1* neurons and absent from *Pet1ⁿᵉᵍ* cells. (**g**) Volcano plot showing the differential chromatin accessibility of TACs between *Pet1⁻/⁻* and wildtype *Pet1* neurons at E14.5, with fold change cutoff of 2-fold and significance cutoff of FDR < 0.01 (top). Highest FDR GO terms for Pet1-dependent TACs (in red) that are occupied by Pet1 show strong enrichment for serotonin biosynthesis. Select genes required for 5-HT synthesis are labeled on the volcano plot. (**h**) The numbers of gain and loss *Pet1⁻/⁻* neuron TACs that map to TACs specific to a subtype of *Pet1* neurons or shared among multiple subtypes in scATAC-seq. (**i–j**) Genome browser tracks showing the gene-enhancer pairings and the ATAC-seq, Pet1 CUT&RUN, and histone modification ChIPmentation signals at 5-HT identity genes. Zoomed-in inserts display the changes in chromatin accessibility between wildtype (red) and *Pet1⁻/⁻* (green) *Pet1* neurons within the highlighted (blue) regions at three developmental time points. Two TACs of the gene *Slc6a4* (red dashed line box) gain accessibility in wildtype 5-HT neurons from E11.5 to E17.5 but fail to open to the same extent in *Pet1⁻/⁻* neurons. Asterisks denote FDR < 0.01 and fold change >2 relative to wildtype at the same time point.

The online version of this article includes the following figure supplement(s) for figure 4:

**Figure supplement 1.** Related to ***Figures 4 and 5***: Pet1 is required for chromatin accessibility at *Pet1* neuron-specific TACs.

*1g*). Indeed, we found a significant loss of accessibility of enhancers either upstream of the TSS or within introns of *Tph2*, *Gch1*, and *Ddc* (***Figure 4i***). TACs associated with the serotonin reuptake transporter genes *Slc22a3* and *Slc6a4* are also significantly reduced in accessibility (***Figure 4j***). Importantly, TACs that require Pet1 for accessibility are mostly unique to *Pet1* neurons, while TACs of 5-HT pathway genes that are not *Pet1* neuron-specific are not significantly affected by Pet1 deficiency (***Figure 4i–j***). These findings indicate that Pet1 controls the chromatin accessibility of *Pet1* neuron-specific TACs of 5-HT neurotransmission genes.

Interestingly, the enhancers associated with *Slc6a4* that are directly occupied by Pet1 show little accessibility to transposase at E11.5 but gradually gain accessibility from E11.5 to E17.5 in wildtype *Pet1* neurons (***Figure 4j***). In contrast, these TAC regions fail to fully open in *Pet1⁻/⁻* neurons (***Figure 4j***), suggesting Pet1 is directly required for initial conversion of these regions to a euchromatic state. Among the Pet1-regulated TACs, we identified 47 regions that similarly fail to fully open between E11.5 and E17.5 in *Pet1⁻/⁻* neurons, 20 of which show direct Pet1 occupancy at E14.5 (***Figure 4— figure supplement 1h-i***). TACs that require Pet1 for chromatin opening are linked to genes enriched for synaptic products, suggesting that Pet1 opens condensed chromatin at TACs controlling synapse functions (***Figure 4—figure supplement 1h***).

## Continued Pet1 function is required to sustain 5-HT regulatory element accessibility

To determine whether early postmitotic neuronal accessible chromatin landscapes require continuous terminal selector input, we analyzed flow sorted E14.5 Pet1 conditional knockout (*Pet1ᶠˡ/ᶠˡ; Pet1-Cre; Rosa26ᴬⁱ⁹*) *Pet1*-lineage neurons ('Pet1-cKO neurons') in which *Pet1* transcript level is maintained for about 2 days till E12.5 (***Liu et al., 2010***; ***Figure 4—figure supplement 1j-l***). We found 1834 Pet1-maintained TACs that are mainly distal CREs in wildtype *Pet1* neurons lose their accessibility in Pet1-cKO (***Figure 5a–d***). The accessibility of 936 of these TACs does not change in wildtype neurons from E11.5 to E14.5, indicating that Pet1 is required to sustain TACs after they open (***Figure 5a***). Many 5-HT neurotransmission gene TACs lose accessibility as in *Pet1⁻/⁻* neurons including the enhancers of 5-HT neurotransmission genes *Tph2*, *Gch1*, *Ddc*, and *Slc22a3* (***Figure 5f–g***).

In addition to selecting 5-HT identity genes for transcriptional activation, Pet1 maintains its expression through positive autoregulation (***Scott et al., 2005a***). CUT&RUN showed that Pet1 directly occupies the TAC that overlaps with its autoregulatory enhancer (***Scott et al., 2005a***; ***Scott et al., 2005b***), which is consistent with our previous Pet1 ChIP-seq findings (***Wyler et al., 2016***; ***Figure 5h***; ***Figure 4—figure supplement 1f***). We found that Pet1 is required for optimal accessibility of this enhancer region and for sustaining its accessibility suggesting that maintenance of chromatin accessibility at autoregulatory enhancers is an important mechanism for terminal selector autoregulation (***Figure 5h***). Together, these findings suggest the early euchromatin landscape in postmitotic 5-HT neurons is a reversible state that requires ongoing Pet1 input for maintenance of accessibility.

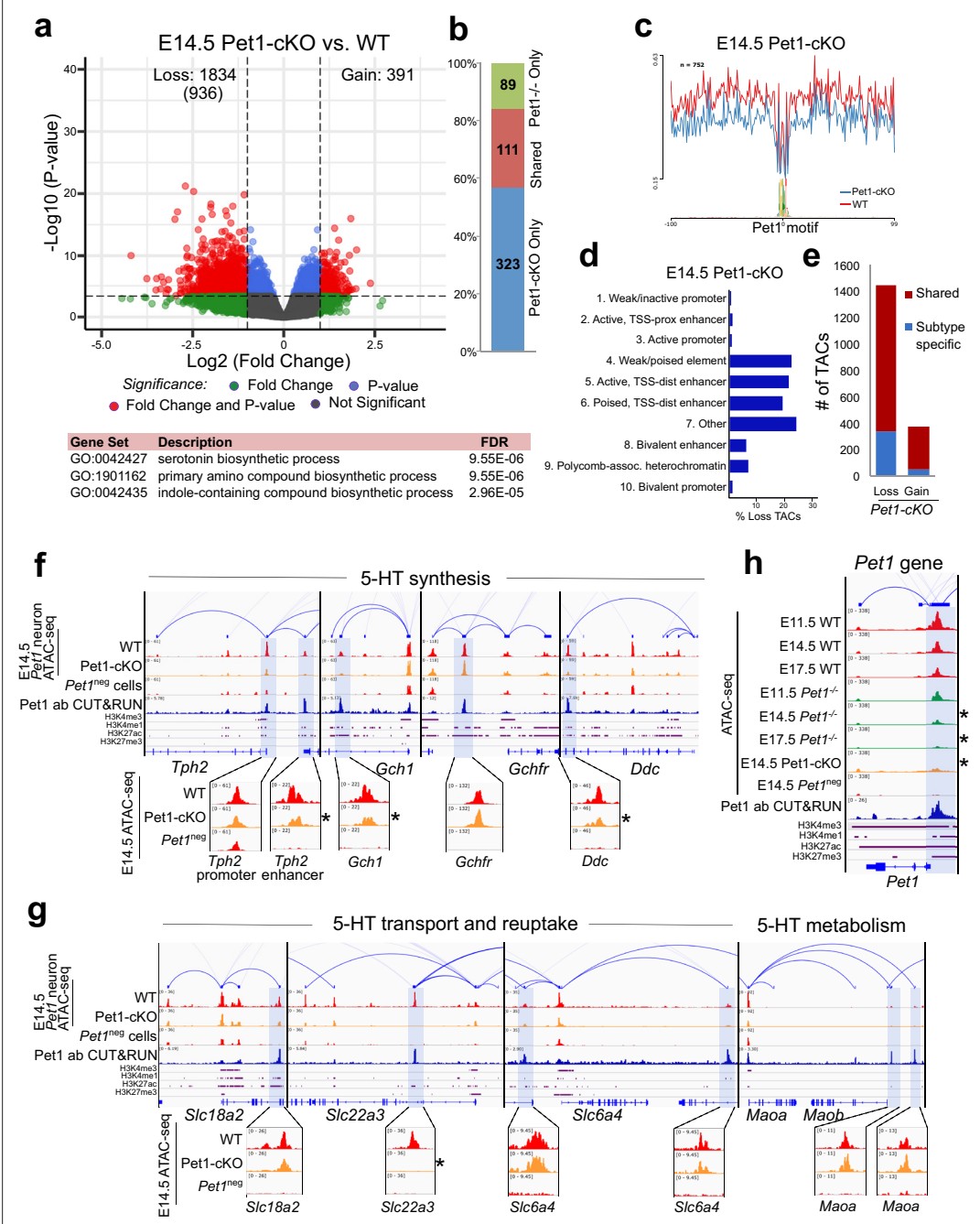

**Figure 5.** Sustained Pet1 input is required for chromatin accessibility. (**a**) Top: Volcano plot showing the differential chromatin accessibility of TACs between Pet1-cKO and wildtype *Pet1* neurons, with fold change cutoff of 2-fold and significance cutoff of FDR < 0.01 (in red). 936 TACs (in parenthesis) show no change in accessibility in wildtype *Pet1* neurons between E11.5 and E14.5 but show loss of accessibility in Pet1-cKO. Bottom: Highest FDR GO terms for the Pet1-regulated TACs (in red) that are occupied by Pet1 show enrichment for 5-HT biosynthesis. (**b**) Overlap between differential TACs in *Pet1⁻/⁻* and Pet1-cKO 5-HT neurons. (**c**) Comparison of in vivo Pet1 footprints over representative ETS-factor motif between Pet1-cKO and wildtype *Pet1* neurons within all differentially accessible chromatin regions. (**d**) The ChromHMM chromatin states genomic coverage for TACs that are maintained by Pet1 at E14.5. (**e**) The numbers of gain and loss *Pet1-cKO Pet1* neuron TACs that map to TACs specific to a subtype of *Pet1* neurons or shared among multiple subtypes in scATAC-seq data. (**f–g**) Genome browser tracks of gene-enhancer pairings, ATAC-seq, Pet1 CUT&RUN, and histone modification ChIPmentation signals at 5-HT identity genes. Zoomed-in inserts display the changes in chromatin accessibility between wildtype (red) and Pet1-cKO (orange) 5-HT precursors within the highlighted (blue) regions at E14.5. Asterisks denote FDR < 0.01 and fold change >2 compared to wildtype. (**h**) Genome browser tracks showing the ATAC-seq, Pet1 CUT&RUN, and histone modification ChIPmentation signals at the *Pet1* locus.

## Lmx1b co-regulates reorganization of chromatin accessibility at 5-HT identity genes

To determine whether reorganization of chromatin accessibility in postmitotic neurons is unique to Pet1, we assessed whether Lmx1b also controls chromatin architecture. Since Lmx1b occupancy in the brain has not been defined, we performed Lmx1b CUT&RUN in E14.5 *Pet1* neurons, which revealed that a significant portion of Pet1 and Lmx1b binding sites coincide within the same TAC regions (*Figure 6—figure supplement 1a-c*). In Lmx1b-cKO (*Lmx1b^fl/fl; Pet1-Cre;Rosa26^Ai9*) *Pet1* neurons (*Donovan et al., 2019*; *Figure 6—figure supplement 1d-f*), chromatin landscape is altered (*Figure 6a*), Lmx1b footprints are reduced (*Figure 6b*), and more TACs are differentially accessible than in Pet1-cKO neurons (*Figures 5a and 6c*), suggesting that Lmx1b has a broader impact than Pet1 on 5-HT neuron chromatin. Lmx1b directly binds 19% of the differentially accessible TACs in Lmx1b-cKO (comparable to Pet1 which binds 20% of differentially accessible TACs in Pet1-cKO) indicating direct maintenance of accessibility by Lmx1b at those regions (*Figure 6d*; *Figure 4—figure supplement 1m*), which are predominantly distal enhancers (*Figure 6e*). Like Pet1-regulated TACs, Lmx1b-regulated TACs are predominantly shared among multiple *Pet1* neuron subtypes (*Figure 6f*). The Lmx1b-dependent TACs with direct Lmx1b occupancy, of which 65% are specific to *Pet1* neurons (*Figure 6i*), are also most significantly associated with monoamine/serotonin synthesis (*Figure 6c*). Moreover, the conditional ablation of Lmx1b results in an even greater closing of 5-HT neurotransmission gene TACs than *Pet1* ablation (*Figure 6j–k*). Many TACs whose accessibility was not significantly reduced in Pet1-cKO neurons (fold change >2, FDR < 0.01) significantly lost accessibility in Lmx1b-cKO neurons (*Figure 6—figure supplement 1j*) including the TACs of *Gchfr*, *Slc18a2*, and *Maoa* (*Figure 6j–k*). This suggests that although Lmx1b is known to activate *Pet1* expression (*Donovan et al., 2019*), the reduction in *Pet1* expression in the Lmx1b-cKO (*Figure 6—figure supplement 1h*) cannot entirely account for all the changes in chromatin accessibility in the Lmx1b-cKO neurons. Lmx1b directly occupies the enhancers of many 5-HT neurotransmission genes (*Figure 6j–k*) whose expression is known to be critically dependent on Lmx1b (*Ding et al., 2003*; *Donovan et al., 2019*).

We also found that the loss of chromatin accessibility in Lmx1b-cKO is highly similar to that in *Pet1* and *Lmx1b* double conditional knockout (DKO: *Lmx1b^fl/fl; Pet1^fl/fl; Pet1-Cre; Rosa26^Ai9*) *Pet1* neurons (*Figure 6b–c , and g–h*), suggesting that the ablation of both TFs does not have greater effect than ablation of Lmx1b alone. PCA also indicates close clustering of Lmx1b-cKO neurons with DKO neurons (*Figure 6—figure supplement 1g*). These findings are consistent with the model that Lmx1b is required for Pet1-dependent chromatin regulation. At a substantial subset of Lmx1b-dependent TACs such as those of *Tph2*, *Gch1*, *Ddc*, and *Slc6a4* (*Figures 5f–g , and 6j–k*), Pet1 and Lmx1b nonredundantly co-regulate the accessibility of the CREs. In parallel, Lmx1b further sustains, independent of Pet1, an additional subset of TACs that are required for axon and synaptic formation, neurotrophin signaling, and neuronal growth (*Figure 6—figure supplement 1i-j*).

Motif analysis on the differentially accessible regions in the cKO and DKO neurons showed that the binding sites of several TF families such as the Rfx-related factors are less accessible in Pet1- and Lmx1b-deficient *Pet1* neurons, demonstrating that together Pet1 and Lmx1b shape the global gene regulatory landscape of available TF binding sites (*Figure 6—figure supplement 1k*).

## Pet1 and Lmx1b regulate 5-HT synapse formation

In addition to controlling TACs associated with 5-HT transmission genes that define serotonergic identity (*Figure 6h*), Pet1 and Lmx1b program TAC accessibility of many genes involved in axon growth and synaptic functions (*Figure 7a*; *Figure 7—figure supplement 1a*). The majority of regulated TACs identified from Lmx1b-cKO and DKO neurons are those that gain accessibility in wildtype *Pet1* neurons from E11.5 to E17.5 (*Figure 7b*). These include the TACs of *Syt1*, which encodes a calcium sensor protein that mediates the release of synaptic vesicles in response to calcium (*Yoshihara and Littleton, 2002*), and *Slc17a8*, which encodes a vesicular glutamate transporter (vGlut3) that enables glutamatergic co-transmission in a subset of adult 5-HT neurons (*Vigneault et al., 2015*; *Figure 7c*). The Pet1- and Lmx1b-dependent TACs of *Syt1* and *Slc17a8* are *Pet1* neuron specific, suggesting that the expression of some widely expressed neuronal genes are regulated in different neurons by neuron-type-specific CREs (*Figure 7c*). In support of the contribution of these Pet1- and Lmx1b-regulated enhancers to developmental gene expression, *Syt1* and *Slc17a8* expression are decreased in Pet1-cKO, Lmx1b-cKO, and DKO neurons (*Figure 7d*). Other axon and synapse-related genes show

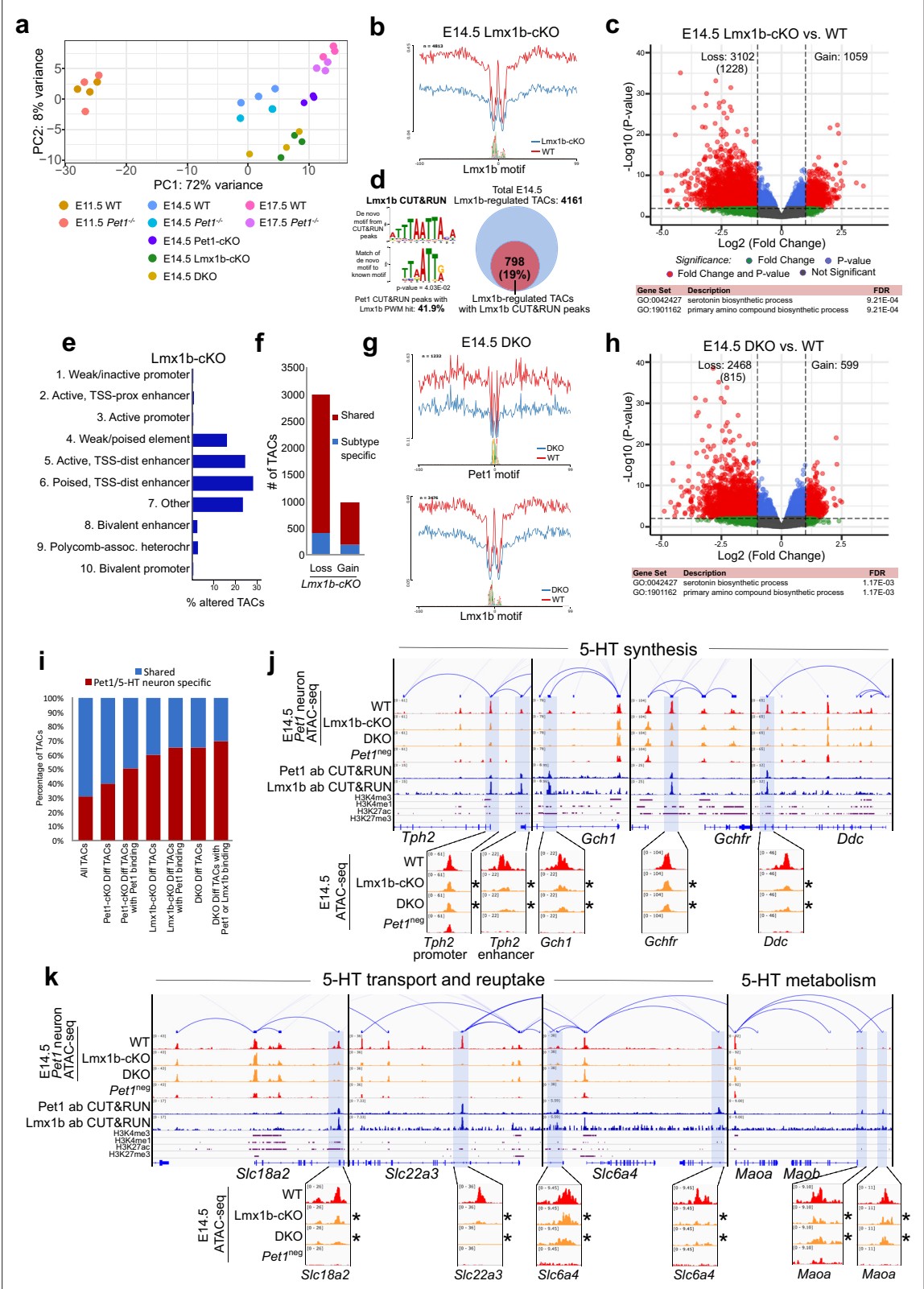

**Figure 6.** Lmx1b co-regulates chromatin accessibility with Pet1 at 5-HT neurotransmission genes but has broader impact on accessibility landscape. (**a**) PCA showing the distribution of wildtype, *Pet1⁻/⁻*, Pet1-cKO, Lmx1b-cKO, and DKO neuron ATAC-seq data over two principal components. (**b**) Comparison of in vivo Lmx1b footprints over representative LIM-HD motif, between Lmx1b-cKO and wildtype *Pet1* neurons within differentially accessible chromatin regions. (**c**) Top: Volcano plot showing the differential chromatin accessibility of TACs between Lmx1b-cKO and wildtype *Pet1*

*Figure 6 continued*

neurons, with fold change cutoff of 2-fold and significance cutoff of FDR < 0.01 (in red). 1228 TACs (in parenthesis) show no change in accessibility in wildtype *Pet1* neurons between E11.5 and E14.5 but show loss of accessibility in Lmx1b-cKO. Bottom: Highest FDR GO terms for Lmx1b-regulated TACs (in red) that are occupied by Lmx1b show enrichment of 5-HT biosynthesis. (**d**) The topmost significantly enriched motif within Pet1 CUT&RUN peaks as determined by de novo MEME motif analysis matches (p = 4.03E-02) the Lmx1b LIM-HD binding site previously defined in vitro (*Jolma et al., 2013*) based on analysis by the TOMTOM Motif Comparison Tool (left panel). Fraction of Lmx1b-maintained TACs containing Lmx1b CUT&RUN peaks (right panel). (**e**) The ChromHMM chromatin states genomic coverage for TACs that lose accessibility in Lmx1b-cKO (blue) and DKO (red) in E14.5 *Pet1* neurons. (**f**) The numbers of gain and loss *Lmx1b-cKO Pet1* neuron TACs that map to TACs specific to a subtype of *Pet1* neurons or shared among multiple subtypes in scATAC-seq data. (**g**) Comparison of the in vivo Pet1 (top) and Lmx1b (bottom) footprints between DKO and wildtype *Pet1* neurons within differentially accessible chromatin regions. (**h**) Top: Volcano plot showing the differential chromatin accessibility of TACs between DKO and wildtype *Pet1* neurons, with fold change cutoff of 2-fold and significance cutoff of FDR < 0.01 (in red). 815 TACs (in parenthesis) show no change in accessibility in wildtype *Pet1* neurons between E11.5 and E14.5 but show loss of accessibility in DKO. Bottom: Highest FDR GO terms for Pet1- and Lmx1b-regulated TACs (in red) that are occupied by the two TFs show enrichment for serotonin biosynthesis. (**i**) The fraction of Pet1-cKO, Lmx1b-cKO, and DKO differentially accessible TACs that is specific to *Pet1* lineage. A significant fraction of the Pet1- and Lmx1b-regulated TACs is unique to *Pet1* neurons and absent from *Pet1*[neg] cells. (**j–k**) Genome browser tracks of gene-enhancer pairings, ATAC-seq, Pet1 CUT&RUN, and histone modification ChIPmentation signals at 5-HT identity genes. Display data ranges for the zoomed-out panels are scaled to the peak in the blue highlighted regions. Zoomed-in inserts display the changes in chromatin accessibility between wildtype (red), Lmx1b-cKO (orange), and DKO (orange) 5-HT precursors within the highlighted (blue) regions at E14.5. Asterisks denote FDR < 0.01 and fold change >2 compared to wildtype.

The online version of this article includes the following figure supplement(s) for figure 6:

**Figure supplement 1.** Pet1 and Lmx1b control the global landscape of TF binding sites.

similar correlation between TAC accessibility and gene expression (*Figure 7—figure supplement 1b-d*). Further, there is significant correlation between TACs that close in Lmx1b-cKO and DKO and the expression of the linked genes (*Figure 7e*). GO analysis of down-regulated genes in DKO neurons revealed that Pet1 and Lmx1b collectively maintain the transcriptome necessary for axon, synapse, and dendrite development (*Figure 7—figure supplement 1b*). Together these data suggest that in addition to controlling accessibility of putative 5-HT transmitter identity enhancers, Pet1 and Lmx1b modulate the accessibility of putative CREs for other 5-HT neuron functions.

To explore the biological significance of Pet1 and Lmx1b-dependent TAC alterations, we analyzed the integrity of 5-HT pericellular baskets that encase postsynaptic neuronal dendrites and cell bodies in the dorsal lateral septum (dLS) (*Senft and Dymecki, 2021*). Lmx1b-cKO 5-HT neurons display axon defects in which most axonal projections fail to reach the dLS and remain stalled within the medial forebrain bundle during development. However, a small number of Lmx1b-cKO 5-HT axons do extend into the dLS where they come in close proximity to post-synaptic cell bodies (*Donovan et al., 2019*). We found that by postnatal day 15 (P15), 5-HT pericellular baskets were readily identified in wildtype brains. Further, as expected, some 5-HT axons reach the dLS in Pet1-cKO, Lmx1b-cKO, and DKO mice (*Figure 7f*). However, pericellular baskets could not be found in the remaining 5-HT axons (*Figure 7f–g*). Of the few axonal terminals that do reach postsynaptic cell bodies, co-localization of Syt1 and vGlut3 immunostaining with TdTomato-positive axon terminals is significantly reduced compared to wildtype animals (*Figure 7h–j*). This suggests that Pet1 and Lmx1b control accessibility of putative CREs for genes that are required for 5-HT axon extension and formation of presynaptic structure that enables 5-HT and glutamatergic co-transmission.

## Discussion

Specialized molecular, morphological, and functional properties of neurons are largely acquired as postmitotic neurons mature (*Polleux et al., 2007*). How postmitotic epigenetic programs control gene expression trajectories to encode mature neuronal identities is poorly understood (*Bradke et al., 2020*; *Gallegos et al., 2018*). Here, we have defined the accessible chromatin landscape and the cis-regulatory architecture of developing *Pet1* neurons that generate mature 5-HT neurons (*Hendricks et al., 1999*). We show that the cis-regulatory landscape of embryonic *Pet1* neurons is highly dynamic as postmitotic *Pet1* neurons acquire serotonergic identities. Our scATAC-seq and scRNA-seq analyses reveal that the chromatin landscapes of neurons derived from a single TF lineage are highly heterogeneous and established early to prefigure and likely encode 5-HT neuron subtypes including those expressing the glutamatergic marker, *Slc17a8*. Previous studies have demonstrated the key role of terminal selectors, Pet1 and Lmx1b, in the differentiation and maturation of 5-HT neurons through

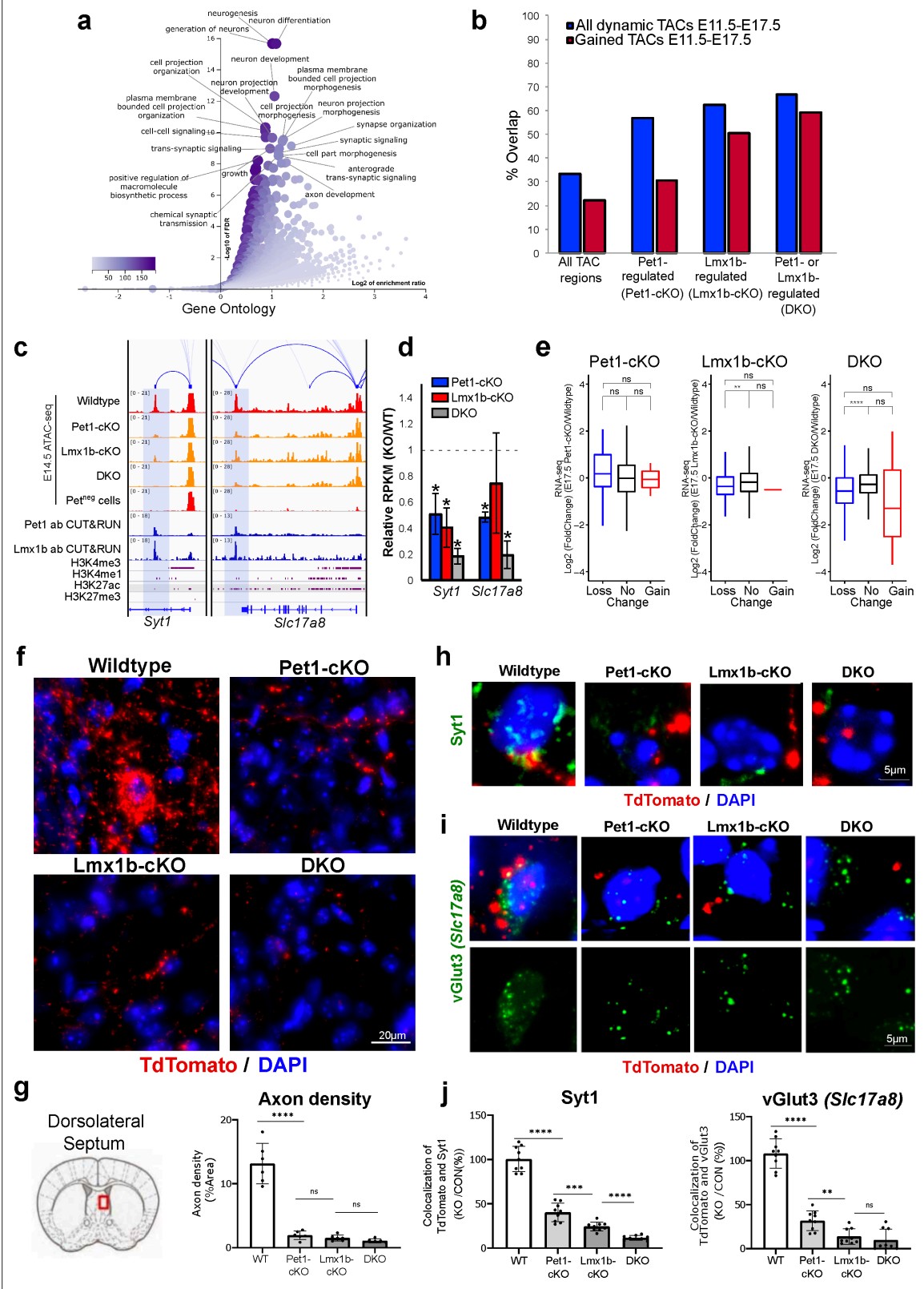

**Figure 7.** Loss of Pet1 and Lmx1b alters chromatin accessibility at synaptic gene enhancers and disrupts 5-HT neuron connectivity. (**a**) GO analysis of all Pet1- and/or Lmx1b-maintained TACs identified from E14.5 DKO ATAC-seq datasets, depicted as volcano plot of enrichment (x-axis) vs. FDR statistical significance (y-axis), showing many synapse and axon biological processes among the top enriched annotations. (**b**) Percent of Pet1- and/or Lmx1b-regulated TACs that gain or lose accessibility (blue) or gain accessibility (red) in wildtype *Pet1* neurons from E11.5 to E17.5. For example, 33% of all TACs

*Figure 7 continued on next page*

*Figure 7 continued*

are dynamic and 22% gain accessibility. In contrast, 62% of Lmx1b-regulated TACs are dynamic and 50% gain accessibility. (**c**) Genome browser tracks of gene-enhancer pairings, ATAC-seq, Pet1 and Lmx1b CUT&RUN, and histone modification ChIPmentation signals at synaptic genes *Syt1* and *Slc17a8*. Tracks show the changes in chromatin accessibility between wildtype (red), Pet1-cKO (yellow), Lmx1b-cKO (yellow), and DKO (yellow) Pet1 neurons within the highlighted (blue) regions at E14.5. Display data ranges for the zoomed-out panels are optimized based on the blue highlighted regions. (**d**) Relative expression (FPKMs) of *Syt1* and *Slc17a8* in rostral Pet1-cKO, Lmx1b-cKO, and DKO neurons at E17.5. Data presented as mean ± SEM. *, FDR<.05. (**e**) Differential expression (from E11.5 to E17.5) of the genes associated with Pet1-occupied TACs in Pet1-cKO vs. wildtype 5-HT neurons (left), genes associated with Pet1-occupied TACs in Lmx1b-cKO vs. wildtype 5-HT neurons (middle), and genes associated with Pet1 and Lmx1b co-occupied TACs in DKO vs. wildtype 5-HT neurons (right). (**f**) TdTomato (red) and DAPI (blue) co-staining of pericellular baskets in the medial septum of P15 *Pet1-Cre; Rosa26^{Ai9}* wildtype, Pet1-cKO, Lmx1b-cKO, and DKO mice. (**g**) Diagram showing the analyzed brain area and quantification of TdTomato axons (pixels/μm²) (two-way ANOVA; n = 3 animals per genotype and 2 quantified brain sections per animal) (bottom panels). Images were generated through automatic stitching of individual 20× images. (**h**) Representative immunohistochemistry images of Syt1, TdTomato, and DAPI in the dorsolateral septum of P15 *Pet1-Cre; Rosa26^{Ai9}* mice showing the decreased overlap of Syt1 (green) and TdTomato (red) signals in KO animals compared to wildtype. Images were generated through automatic stitching of individual 63× images. (**i**) Representative immunohistochemistry images of vGlut3 (*Slc17a8*), TdTomato, and DAPI in the medial septum of adult *Pet1-Cre; Rosa26^{Ai9}* mice showing the decreased overlap of vGlut3 (green) and TdTomato (red) signals in KO animals compared to wildtype. Images were generated through automatic stitching of individual 63× images. (**j**) Quantification of Syt1 (left) and vGlut3 (right) co-localization with TdTomato in the medial septum as a percentage of the total TdTomato signal (two-way ANOVA; n = 2 animals per genotype and 3–4 quantified images per animal). *, P<.05; **, P<.01; ***, P<.001; ****, P<.0001 in panels e, g, j.

The online version of this article includes the following figure supplement(s) for figure 7:

**Figure supplement 1.** Pet1 and Lmx1b control chromatin access to synapse and axon gene linked TACs.

sequence-specific transactivation of serotonergic terminal effector genes. We show here that an additional functional attribute of Pet1 and Lmx1b is to reorganize postmitotic chromatin accessibility to enable sequence-specific activation of serotonergic neurotransmission and synapse genes. Thus, our findings identify developmental regulators of postmitotic neuronal chromatin accessibility.

We found that distal transcriptional CRE accessibility is heterogeneous not only between *Pet1* neurons and other cell types but also within the *Pet1*-lineage neurons. Diversity in regulatory sequence accessibility is predictive of the transcriptomic profile of single *Pet1* neurons. Furthermore, the maturation of *Pet1* neuron putative CRE accessibility is highly correlated with the expression trajectories of their target genes as serotonergic functional properties progressively emerge. TACs linked to genes involved in synapse organization and axon development gain accessibility while those associated with cell cycle control and early embryonic morphogenesis lose accessibility. Importantly, the chromatin landscape is more dynamic between E11.5 and E14.5 than between E14.5 and E17.5, suggesting that the early postmitotic period comprises a highly active stage of chromatin maturation. Since *Pet1* neurons have exited the cell cycle, the temporal changes in open chromatin cannot be explained as an altered abundance of *Pet1* neuron subtypes. Rather, chromatin landscape within individual *Pet1* neurons is dynamic during postmitotic maturation.

The regulatory factors controlling chromatin dynamics in postmitotic neurons are not well defined. TFs with known pioneer activity such as Ascl1 that are expressed in serotonergic progenitors are significantly downregulated or undetectable in postmitotic *Pet1* neurons (*Aydin et al., 2019*; *Wyler et al., 2016*). Yet, our data suggests that new postmitotic euchromatic regions dynamically appear while other euchromatic regions are lost as 5-HT neuron identity is being acquired, suggesting the existence of regulatory factors that shape postmitotic neuronal chromatin. Additionally, several recent studies computationally mined the genome-wide accessible chromatin sequences and identified TF binding sites that are enriched in regions of dynamic chromatin in neurons or brain regions (*Gray et al., 2017*; *Preissl et al., 2018*; *de la Torre-Ubieta et al., 2018*; *Trevino et al., 2021*).

Our findings support a founding paradigm in which terminal selector TFs, at least Pet1 and Lmx1b in mouse 5-HT neurons, reorganize the postmitotic neuronal epigenome to select specific gene CREs. We show that targeting of *Pet1* and *Lmx1b* in the early stages of postmitotic maturation significantly alters the *Pet1* neuron accessible chromatin landscape. CUT&RUN assays suggest that Pet1 and Lmx1b directly control the accessibility of 5-HT-specific CREs linked to genes encoding terminal effectors of 5-HT identity and neurotransmission. In fact, we provide evidence in support of a role for Pet1 in initiating the conversion to euchromatin of multiple nucleosome-protected cis-regulatory sequences in postmitotic neurons, including enhancers linked to the serotonin transporter gene, *Slc6a4*. We further show that Pet1 is required for accessibility of its own autoregulated enhancer region that directs 5-HT neuron expression. Autoregulation is thought to boost and maintain terminal selector expression

(*Leyva-Díaz and Hobert, 2019*). We found that in the absence of *Pet1* expression, *Pet1* enhancer accessibility is significantly reduced but is not fully closed. This suggests that Pet1 boosts its own expression through augmenting auto-enhancer accessibility that was previously initiated by upstream pioneers but was insufficient for proper expression levels of Pet1.

We present evidence suggesting Pet1 and Lmx1b are required to maintain 5-HT-specific open chromatin regions. Whereas pioneer factors in progenitor cells can drive cell lineage commitment and induce, through transient input, stable 'memories' of epigenetic states that are transmitted through further cell divisions (*D'Urso and Brickner, 2014*), our findings suggest that postmitotic accessible chromatin landscapes are unstable and require continuous terminal selector input at least during early postmitotic maturation to sustain the euchromatin state of CREs controlling terminal effector genes. Surprisingly, a larger number of chromatin accessibility changes were found after *Pet1* conditional targeting compared to constitutive germline *Pet1* targeting. We do not have a straightforward explanation for this difference but it may be due to compensatory effects from other regulators. While Lmx1b maintains accessibility of a distinct set of CREs independent of Pet1, chromatin accessibility of many *Pet1*-lineage TACs requires direct and non-redundant co-regulation by both TFs, reminiscent of the cooperative activities of TFs and pioneer factors in other cell types (*Holmberg and Perlmann, 2012*; *Mazzoni et al., 2013*; *Zaret and Carroll, 2011*). Because a large number of cognate motif sites can exist in the genome for each individual TF, selective chromatin accessibility controlled by collaborative postmitotic reorganizers may serve to ensure that neuron-type-specific effector genes are selectively activated only in the presence of multiple positive regulatory inputs. Other TFs may also perform similar function in other neuron types (*Allaway et al., 2021*). The mechanism by which Pet1 and Lmx1b program neuronal chromatin is not understood, but could involve the recruitment of chromatin remodelers or direct cooperative displacement of histones (*Spitz and Furlong, 2012*).

In addition to neurotransmitter synthesis, neurons must develop axonal and synaptic connectivity in order to transmit signals (*Kratsios et al., 2015*; *Spencer and Deneris, 2017*). Pet1 and Lmx1b reorganize the access to CREs for both 5-HT neurotransmission and synaptic connectivity. As a result, the acquisition of distinct functional properties is developmentally coupled. A growing body of evidence supports the idea that alterations in 5-HT neurotransmission cause behavioral changes related to those that characterize neuropsychiatric disorders such as anxiety and autism (*Doan et al., 2019*; *Suri et al., 2015*). As many of these disorders are neurodevelopmental in origin, the study of the chromatin regulatory programs that control 5-HT neuron and other identities may shed light on the pathophysiology of brain disorders at critical stages of brain development.

## Materials and methods
### Animals
Mice were maintained in ventilated cages on a 12 hr light/dark cycle with access to food and water ad libitum with two to five mice per cage. The transgenic mouse lines used in this study have been described previously (*Donovan et al., 2019*; *Wyler et al., 2016*). Conditional knockout mice (Lmx1b-cKO, Pet1-cKO, DKO) were generated by breeding animals bearing *Rosa26*^tdTomato^ (Ai9; Jackson Labs Stock #: 007909) and either the *Lmx1b*^fl/fl^ (*Zhao et al., 2006*), *Pet1*^fl/fl^ (*Liu et al., 2010*), or both alleles with animals harboring *Pet1-Cre* (original name ePet-Cre) transgene. *Pet1*^+/+^; *Pet1-EYFP* and *Pet1*^-/-^; *Pet1-EYFP* mice were generated by crossing *Pet1*^+/-^ bearing the *Pet1-EYFP* transgene. The *Pet1-EYFP* transgene expresses Yfp under the same *Pet1* enhancer as the *Pet1* enhancer region driving the *Pet1-Cre*. Genotyping was performed using ear tissues with the following primers: Pet-1: 5'-CGGTGGATGTGGAATGTGTGCG-3', 5'-CGCACTTGGGGGGGTCATTATCAC-3', 5'-GCCT GATGTTCAAGGAAGACCTCGG-3'; floxed Pet-1: 5'-TAGGAGGGTCTGGTGTCTGG-3', 5'-GCGT CCTTGTGTGTAGCAGA-3'; Pet-EYFP: 5'-TATATCATGGCCGACAAGCAG-3', 5'-GAACTCCAGCAG GACCATGT-3'; floxed Lmx1b: 5'- AGGCTCCATCCATTCTTCTC-3', 5'- AGGCTCCATCCATTCTTCTC -3'; Rosa26^tdTomato^: 5'-CCACAATAAGCAAGAGGCAC-3', 5'- CCGAAAATCTGTGGGAAG TC-3', 5'- CCGAAAATCTGTGGGAAG TC-3', 5'- CTGTTCCTGTACGGCATG G-3'; Pet1-Cre: 5'- CGGCATCA ACGTTTTCTTTT –3', 5'- AGTCAGGGCAGAGCCATCTA –3'.

Embryonic age was defined as the number of days following the appearance of a vaginal plug, designated as embryonic day (E) 0.5. For all procedures, animals were euthanized by anesthesia with isoflurane inhalation followed by cervical dislocation, and brain tissues were collected equally from

male and female mice or embryos. All animal experiments were approved by the Institutional Animal Care and Use Committee of CWRU.

## Neuron dissociation and flow cytometry

Isolation of *Pet1* neurons was performed as previously described with a few modifications (*Donovan et al., 2019*). Briefly, E11.5 or E14.5 embryos from *Pet1-Cre⁺; Rosa26^Ai9* or *Pet1-EYFP* animals were dissected in cold PBS under dissection microscope to collect all fluorescent cells in the fetal mouse hindbrain. Dissected tissues were transferred into cold 1× TrypLE Express (Gibco) and incubated at 37°C for 15 min with gentle agitation. Following enzymatic trypsin digestion, samples were centrifuged for 1 min at 1500 rpm, TrypLE Express was removed, and samples were washed and resuspended in 500 µL cold Leibovitz's L-15 media (Thermo Fisher) with 4 µL DNAse I (Invitrogen) and 10% FBS (Gibco). Cells were then slowly triturated with fire-polished glass Pasteur pipettes of decreasing bore size until a single-cell suspension was achieved.

E17.5 embryos were dissociated using a modified adult neuron dissociation method. *Pet1-Cre⁺; Rosa26^Ai9* or *Pet1-EYFP* embryos were dissected in cold aCSF solution (3.5 mM KCl, 126 mM NaCl, 20 mM NaHCO₃, 20 µM dextrose, 1.25 mM NaH₂PO₄, 2 mM CaCl₂, 2 mM MgCl₂, 50 µM AP-V (Tocris), 29 µM DNQX (Sigma), and 100 nM TTX (Abcam)) to collect all fluorescent cells in the hindbrain. Dissected tissues were then incubated in bubbling (95% O₂, 5% CO₂) aCSF containing 1 mg/mL protease from *Streptomyces griseus* (Sigma) for 15 min at room temperature. Following enzymatic digestion, tissues were transferred to 500 µL cold aCSF/10% FBS solution containing 4 µL DNase I (Invitrogen) and gently triturated with fire-polished glass pipettes to generate single-cell suspension.

Dissociated cells were filtered through a 40 µm filter (BD Biosciences) and sorted on a Becton Dickinson FACS Aria or FACS Aria-SORP digital cell sorter with 85 µm nozzle. Viability of flow sorted cells were routinely checked with Trypan blue staining followed by manual counting of intact cells on small portions of the FACS-purified neurons.

## Bulk *Pet1* neuron ATAC-seq library preparation

ATAC-seq was carried out as previously described (*Buenrostro et al., 2015*) with a few modifications using rostral and caudal *Pet1* neurons flow sorted with either *Pet1-EYFP* (wildtype and *Pet1⁻/⁻* samples) or *Pet1-Cre; Rosa26^Ai9* (Pet1-cKO, Lmx1b-cKO, and DKO samples). Bulk ATAC-seq for each time point/condition was performed with three to five biological replicates. Each biological replicate consisted of embryos flow sorted on the same day from either one or multiple dams that were then pooled to collect a total of 10,000–50,000 *Pet1* neurons. Neurons were sorted into L15/10% FBS solution, washed with ice-cold PBS, and resuspended in 25 µL ice-cold ATAC-RSB buffer (10 mM Tris-HCl, pH 7.4, 10 mM NaCl, 3 mM MgCl₂, 0.1% NP40 [Roche cat#11332473001]). Cells were incubated in ATAC-RSB on ice for 2 min, then centrifuged at 800× *g* for 10 min at 4°C. Pelleted nuclei were resuspended in 25 µL Transposition mix (12.5 µL 2× TD buffer [Illumina], 1.25 µL transposase [100 nM final] [Illumina], 8.25 µL PBS, 3.25 µL H₂O) for E11.5 and E14.5 samples and 15 µL of Transposition mix for E17.5 samples. The transposition reactions were incubated at 37°C for 25 min on a Thermomixer at 1000 rpm. The resulting DNA fragments were purified using the DNA Clean and Concentrator-5 Kit (Zymo) and PCR amplified for 13 cycles with Illumina Nextera adapter primers using the NEBNext High Fidelity 2× Master Mix (NEB) with the following PCR program: (1) 5 min at 72°C, (2) 30 s at 98°C, (3) 10 s at 98°C, (4) 30 s at 63°C, (5) 1 min at 72°C, and (6) repeat steps 3–5, 12×. Final PCR products were cleaned using PCRClean Dx beads (Aline Biosciences), assessed for quality on a Bioanalyzer, and sequenced for 75 cycles to an average depth of 29 M paired-end reads per sample on an Illumina NextSeq 550.

## ChIP-seq using Tagmentation (ChIPmentation)

*Pet1* neurons (caudal and rostral) were flow sorted from *Pet1-EYFP* embryos at E14.5 into PBS/10% FBS. Neurons were frozen in PBS as cell pellets in –80°C until use; 50,000–300,000 frozen cells from multiple days were then thawed on ice, fixed with 1% formaldehyde in PBS for 10 min at room temperature followed by quenching with 0.125 M glycine, and pooled to generate one biological replicate. Two biological replicates were collected for each antibody. ChIP-seq using Tagmentation (ChIPmentation) was then performed as previously described (*Schmidl et al., 2015*). Cells were washed once with ice-cold PBS and lysed in 130 µL Sonication Buffer (10 mM Tris-HCl pH 8, 0.25% SDS, 2 mM

EDTA, 1× protease inhibitor cocktail [Active Motif # 37490], 1 mM PMSF). Sonication was conducted in MicroTUBEs (130 µL) on a Covaris S2 ultrasonicator until DNA fragments were in the range of 200–700 bp. Lysates were diluted 1:1.5 with Equilibrium Buffer (10 mM Tris-HCl pH 8, 233 mM NaCl, 1.66% Triton X-100, 0.166% sodium deoxycholate, 1 mM EDTA, 1× protease inhibitor cocktail [Active Motif]) and incubated with 3 µg of antibody on a rotator overnight at 4°C. The following antibodies were used: H3K4me3 (Diagenode C15410003-50), H3K4me1 (Abcam ab8895), H3K27ac (Abcam ab4729), H3K27me3 (Active Motif 39055). The next day, 10 µL of Protein A/G Dynabeads blocked overnight with 0.1% BSA in RIPA-LS Buffer (10 mM Tris-HCl pH 8, 140 mM NaCl, 1 mM EDTA, 0.1% SDS, 0.1% sodium deoxycholate, 1% Triton X-100) were transferred to each immunoprecipitated lysate and incubated for 2 hr at 4°C on a rotator. Following incubation, beads were washed twice with ice-cold RIPA-LS Buffer, twice with cold RIPA-HS Buffer (10 mM Tris-HCl pH 8, 500 mM NaCl, 1 mM EDTA, 0.1% SDS, 0.1% sodium deoxycholate, 1% Triton X-100), twice with RIPA-LiCl Buffer (10 mM Tris-HCl pH 8, 1 mM EDTA, 250 mM LiCl, 0.5% NP-40, 0.5% sodium deoxycholate), and twice with 10 mM Tris-HCl pH 8. Beads were then resuspended in 25 µL of tagmentation reaction mix containing 12.5 µL of 2× Tagmentation DNA Buffer (Illumina #15027866) and 1 µL of Tagment DNA Enzyme (Illumina #15027865). Tagmentation reaction was incubated at 37°C for 10 min with gentle vortex every 5 min. Tn5 reaction was inactivated by adding 500 µL of ice-cold RIPA-LS Buffer. Beads with washed twice with RIPA-LS Buffer and twice with TE Buffer (10 mM Tris-HCl pH 8, 1 mM EDTA) and resuspended in 48 µL of ChIP Elution Buffer (10 mM Tris-HCl pH 8, 5 mM EDTA, 300 mM NaCl, 0.4% SDS) and 2 µL of Proteinase K solution (Thermo Fisher). Digestion occurred at 55°C for 1 hr followed by sample de-crosslinking at 65°C overnight. Supernatant was transferred the next day to a new tube, and Proteinase K digestion was repeated on remaining bead-bound chromatin by resuspending the beads in 1 µL Proteinase K in 19 µL ChIP elution buffer and eluting for 1 hr at 55°C. Supernatants from both elutions were combined and purified using DNA Clean and Concentrator-5 Kit (Zymo). Library preparation was performed as described for ATAC-seq. ChIPmentation libraries were analyzed on a Bioanalyzer for quality control and then sequenced with 2 × 75 paired-end reads per sample on an Illumina NextSeq 550.

## scATAC-seq library construction

*Pet1* neurons from *Pet1-EYFP* embryos were flow sorted at E14.5 into PBS/10% FBS, washed twice with PBS/0.04% BSA, and resuspended in 100 µL freshly prepared Lysis Buffer (10 mM Tris-HCl pH 7.4, 10 mM NaCl, 3 mM MgCl$_2$, 0.1% Tween-20, 0.1% IGEPAL). After 3 min of cell lysis on ice, 1 mL of chilled Wash Buffer (10 mM Tris-HCl pH 7.4, 10 mM NaCl, 3 mM MgCl$_2$, 1% BSA, 0.1% Tween-20) was added to the reaction and nuclei were pelleted and resuspended in 1× Nuclei Buffer (10× Genomics). Nuclei concentration and quality were determined using Trypan blue staining followed by cell counting and inspection on a hemocytometer to ensure that nuclei were free of nuclear membrane blebbing and that there was complete lysis. Nuclei were then added to a transposition mix to perform the Tn5 transposase reaction. The reaction was incubated at 37°C for 60 min. Transposed nuclei were then loaded onto the 10× Genomics Chromium controller to perform nuclei encapsulation with barcoded gel beads, followed by library amplification for 13 cycles. After purification, the library was analyzed on an Agilent Fragment Analyzer to determine fragment size. Library was sequenced on Illumina NextSeq 550 Sequencing System to a depth sufficient to yield >50,000 paired-end reads per nuclei.

## scRNA-seq library construction

*Pet1* neurons from *Pet1-EYFP* embryos were flow sorted at E14.5 into PBS/10% FBS. FACS-purified neuron viability was confirmed to be >80% by staining a portion of the cells with Trypan blue and counting the number of live cells on a hemocytometer. Single cells were loaded in the 10× Single Cell 3' Chip at a concentration of 1000 cells/µL. Library was prepared according to 10× Genomics Single Cell Protocols Cell Preparation Guide on the 10× Genomics Chromium controller and sequenced to a depth of over 80,000 reads per cell for ~2700 cells on Illumina NextSeq 550 Sequencing System.

## CUT&RUN library preparation

CUT&RUN was performed based on the Epicypher protocol v1.6, which is derived from the method developed by *Skene et al., 2018*. Briefly, Concanavalin A-coated (ConA) beads were washed twice with CUT&RUN Bead Activation Buffer (20 mM HEPES pH 7.9, 10 mM KCl, 1 mM CaCl$_2$, 1 mM MnCl$_2$).

FACS-purified *Pet1* neurons were washed twice with Wash Buffer (20 mM HEPES pH 7.5, 150 mM NaCl, 0.5 mM spermidine, protease inhibitors [Roche]) and then incubated with activated ConA beads for 15 min at room temperature to allow the cells to absorb to the beads. ConA bead-bound cells were incubated overnight in 50 µL Antibody Buffer (Wash Buffer, 0.01% digitonin, 2 mM EDTA) with 1:100 rabbit anti-FEV (Sigma-Aldrich #HPA067679), guinea pig anti-Lmx1b (Carmen Birchmeier, Max Delbruck Center Berlin), or rabbit monoclonal IgG antibody (Jackson ImmunoResearch Lab 115-007-003). One replicate was collected for each antibody. After overnight incubation, the ConA-bound cells were washed twice with Digitonin Wash Buffer (Wash Buffer, 0.01% digitonin), resuspended in 50 µL Digitonin Buffer containing 700 ng/mL CUTANA pAG-MNase (Epicypher), and incubated for 10 min at room temperature. After binding of pAG-MNase, ConA bead-bound cells were washed twice with Digitonin Buffer, then resuspended in 50 µL Digitonin Buffer containing 2 mM $CaCl_2$ and incubated at 4°C for 2 hr on a rotator to induce digestion of the chromatin. Reaction was stopped by the addition of 33 µL STOP Buffer to each sample (340 mM NaCl, 20 mM EDTA, 4 mM EGTA, 50 µg/mL RNase A [Thermo Fisher], 50 µg/mL GlycoBlue [Thermo Fisher]) and incubated at 37°C for 15 min to release cleaved chromatin into the supernatant and degrade RNA. DNA was purified from the supernatant using the DNA Clean and Concentrator-5 kit (Zymo). Sequencing library preparation was performed using the NEBNext Ultra II Library Prep Kit for Illumina (NEB) according to the manufacturer's instructions, with the following modification: following adapter ligation DNA was purified using 1.75× PCRClean Dx beads (Aline Biosciences) and libraries were amplified using the following CUT&RUN-specific PCR cycling conditions to enrich for 100–700 bp fragments: (1) 45 s at 98°C, (2) 15 s at 98°C, (3) 10 s at 60°C, (4) repeat steps 2–3 for a total of 14×, and (5) 1 min at 72°C. PCRClean Dx beads were used to purify the final libraries, which were then sequenced at 75bp x2 on an Illumina NextSeq 550.

## RNA-seq library preparation

Rostral *Pet1* neurons were dissected from E17.5 DKO (*Pet1-Cre; Pet1^{fl/fl}; Lmx1b^{fl/fl}; Rosa26^{Ai9}*) or control *Pet1-Cre; Pet1^{+/+}; Lmx1b^{+/+}; Rosa26^{Ai9}* embryos and flow sorted into Trizol LS (Invitrogen) on a Becton Dickonson FACS Aria-SORP sorter. Experiment was performed with three control and two DKO biological replicates. Each biological replicate represented 1000–3000 pool flow sorted neurons from multiple embryos of one or more litters. Total RNA was isolated using chloroform extraction and purified using RNA Clean & Concentrator-5 Kit (Zymo). RNA concentration and quality were measured using Quantifluor RNA system (Promega) and Agilent Fragment Analyzer (Advanced Analytics). DNase I treatment and library construction was performed using the Ovation SoLo RNA-Seq Systems V2 according to manufacturer's instructions. Final libraries were sequenced on a NextSeq 550 (Illumina) with single-end sequencing for 75 cycles.

## Immunohistochemistry, image acquisition, and processing

Mice were deeply anesthetized by isoflurane inhalation (chamber atmosphere containing 4% isoflurane) and perfused for 20 min with cold 4% paraformaldehyde (PFA) in PBS. Brains were post-fixed in 4% PFA overnight and transferred to 30% sucrose/PBS overnight. Tissues were then frozen in optimal cutting temperature solution and sectioned on a cryostat at 25 µm thickness. Tissue sections were mounted on SuperFrost Plus slides (Thermo Fisher Scientific) and dried. For antigen retrieval, sections were heated up in sodium citrate buffer for 5 min in the microwave at low power. Sections were blocked with 10% NGS in PBS-T for 1 hr, followed by blocking of endogenous biotin with Avidin/Biotin Blocking kit (Vector) only for Syt1 and vGlut3 stained samples. Sections were incubated with primary antibody (1:1000) at room temperature overnight, then treated the next day with secondary antibodies for 1 hr, followed by visualization (Tph2, GFP, and RFP) or further processing with the VECTASTAIN ABC kit (VECTOR) according to manufacturer's instructions before visualization. Primary antibodies used were: mouse anti-vGlut3 antibody (Abcam ab134310), mouse anti-Synaptotagmin1 (Sigma, SAB1404433), rabbit anti-Tph2 (Millipore Sigma, ABN60), rabbit anti-RFP (Rockland 600-401-379), and chicken anti-GFP (Abcam, ab13970). Secondary antibodies used were: goat anti-mouse, Alexa Fluor 488 or 647 (Invitrogen, A10029 or A28181; 1:500), goat anti-chicken, Alexa Fluor 488 (Invitrogen, A11039, 1:500), goat anti-rabbit, Alexa Fluor 594 (Invitrogen, A11037, 1:500), and biotinylated goat anti-guinea pig IgG antibody (Vector, BA7000; 1:200).

Stained slides were imaged on an LSM800 confocal microscope (Carl Zeiss). In each experiment two animals are analyzed per genotype. Images were acquired per animal using 20× and 63× objective lens and processed in ImageJ. Signal brightness and contrast were edited across whole images equally among genotypes for all pericellular basket synapse and axon images. Signal intensity calculations were normalized for each section. Two-way ANOVA with Welch's correction analysis was performed.

## ATAC-seq data processing and analysis

ATAC-seq data were processed using the standardized software pipeline from the ENCODE consortium to perform quality-control, read alignment, and peak calling (*Landt et al., 2012*). FASTQ files from ATAC-seq reads were mapped to UCSC mm10 with Bowtie v2.3.4.3. Samtools v1.9 was used to remove PCR duplicates and chrM reads. Peak calling was performed using MACS2 with the following parameters: --nomodel --shift 37 --ext 73 --pval 1e-2 -B --SPMR --callsummits. Then by intersecting called peaks in replicates using BEDTools v2.29.0, we defined replicated peaks as the set of peaks present independently in each replicated peak call set and also called in the pooled set.

## Differential accessibility analysis

A comprehensive set of ATAC-seq peaks (TACs) from control datasets (E11.5, E14.5, and E17.5) was generated to be used as the reference for assessing differential accessibility between control time points and for control to TF knockout comparisons. All control peak sets were merged using 1 bp overlap to create the union of all control peaks (E11.5, E14.5, and E17.5). Any potential batch effects were corrected using surrogate variable analysis (*Leek, 2014*). The number of significant surrogate variables was determined using an iteratively re-weighted least squares approach. One significant surrogate variable was found and added to the model design in DESeq2 (*Love et al., 2014*). Differentially accessible peaks were determined using DESeq2 v1.32.0 with a fold change cutoff of 2-fold and FDR adjusted p-value ≤ 0.01 for all wildtype and conditional knockout comparisons and ≤0.05 for *Pet1*[-/-] comparisons. BEDTools v2.30.0 was used to find shared intersections between identified peaks (1 bp minimum overlap). Peak annotations were performed using ChIPseeker v1.28.3. PCA was performed using the prcomp function within the 'plotPCA' wrapper in DESeq2 on variance stabilizing transformed count data to correct for the underlying mean-variance relationship and plotted with ggplot2 (*Love et al., 2014*).

Volcano plots were generated using the EnhancedVolcano package in R (v1.10.0, https://github.com/kevinblighe/EnhancedVolcano; *Blighe, 2021*).

## Motif enrichment

TF motifs were identified using monaLisa v0.1.50 (https://fmicompbio.github.io/monaLisa/index; *Turaga, 2022*). Differentially accessible peak regions were binned into 'loss', 'no change', and 'gain' categories and motif enrichment for each bin was compared against an equal number of sequences randomly sampled from the genome matched by GC composition. Statistically significant enriched motifs were determined using a one-tailed Fisher's exact test with Benjamini-Hochberg adjusted p-values ≤ 1e-8.

## GO enrichment

GO was performed using Webgestalt requiring >5 and <2000 genes per category and FDR ≤ 0.05 (*Liao et al., 2019*).

## ChIPmentation data processing and analysis

ChIPmentation data were processed using the standard ENCODE ChIP-seq software pipeline from the ENCODE Consortium. Sequencing reads were mapped to the UCSC mm10 by Bowtie v2.3.4.3. PCR duplicates were removed using samtools v1.9. The unique mapped reads were used to call peaks comparing immunoprecipitated chromatin with input chromatin with MACS2 v2.2.4 with FDR adjusted p-value cutoff of 0.01. Peak lists were filtered to remove peaks overlapping ENCODE blacklisted regions. BEDTools was used to find shared intersections between identified peaks (1 bp minimum overlap). Peak annotations were performed using ChIPseeker v1.28.3.

## Chromatin state analysis

Chromatin state analysis was performed using ChromHMM v1.22 (*Ernst and Kellis, 2017*). Four histone mark ChIPmentation datasets were binarized at a resolution of 200 bp with a signal threshold of 4. We trained a 10-state model as it characterized the key interactions between the histone marks. The posterior probability state of each genomic bin was calculated using the learned model.

## Super-enhancer analysis

The rank ordering of super-enhancers (ROSE) package was used to identify super-enhancers based on the ranking of H3K27ac signal intensities from E14.5 *Pet1* neurons (*Lovén et al., 2013*; *Whyte et al., 2013*). Clusters of E14.5 ATAC-seq peaks within 12.5 kb were grouped and ranked for their total input background subtracted H3K27ac signal with super-enhancers called above the inflection point of 6391.9.

## scATAC-seq analysis

Raw scATAC-seq data was processed using the CellRanger ATAC v1.2.0 pipeline (10× Genomics). Reads were aligned to the GRCm38 (mm10) mouse genome. Peaks were called using MACS2 v2.2.4. The peak/cell matrix and all unique single-cell fragments were imported into R using the Signac package (*Stuart et al., 2021*). Several QC metrics were calculated to assess the quality of the data. Cells with >3000 to <100,000 peak region fragments, >40% reads in peaks, <0.025 blacklist ratio, <4 nucleosome signal, >2 TSS enrichment were retained for further analysis. The data was normalized and dimensionality reduced using latent semantic indexing (*Cusanovich et al., 2015*). First, term frequency-inverse document frequency (TF-IDF) normalizes across cells for sequencing depth and across peaks to give higher values to rare peaks. Next, we ran singular value decomposition on the TF-IDF matrix using all variable features. A gene activity matrix was constructed by counting the number of fragments that map within the 2 kb upstream region of each gene. Motif enrichment and TF activity for scATAC-seq clusters were quantified using the 'RunChromVAR' wrapper function within the Signac package (v1.14.0) (*Schep et al., 2017*; *Stuart et al., 2021*). The filtered cell (n = 1692) by peak (n = 124,111) matrix was used as input with the curated database of mm10 cisBP motifs (n = 884). Default settings were used to compute motif accessibility deviation Z-scores.

## scRNA-seq analysis

Raw scRNA-seq data was processed using the CellRanger v3.1.0 pipeline (10× Genomics). Reads were aligned to the GRCm38 (mm10) mouse genome. The unique molecular index count matrix was imported into R using Seurat (*Butler et al., 2018*; *Hao et al., 2021*; *Satija et al., 2015*; *Stuart et al., 2019*). Cells with <200 or >7500 expressed genes or >10% mitochondrial genes were excluded from further analysis. Gene counts were log normalized, then the dataset was scaled and centered and mitochondrial gene levels were regressed out. Dimensionality of the data was determined using an elbow plot and the first 20 principal components were used to reduce dimensionality using PCA.

## Integration of scATAC-seq and scRNA-seq data

scATAC-seq and scRNA-seq datasets were integrated using SCTransform from Seurat using 3000 variable features. To perform clustering, we first computed a PCA on the normalized count matrix and used the first 20 PCs to generate a shared nearest neighbor graph with the k parameter set to 20. Then, we clustered the data with a resolution of 0.8 using the Louvain algorithm. Next, we performed UMAP to project the clustering onto low dimensional space and plot the first two dimensions for visualizations (30 nearest neighbors, minimum distance = 0.3) (*McInnes et al., 2018*). Cluster markers were determined using the Wilcoxon rank sum test with a minimum fraction of cells expressing the gene at 0.25 and a log fold change threshold of 0.25 for each cluster vs. all other clusters. The top two marker genes for each cluster are used to identify each cluster.

## Adult scRNA-seq projection onto embryonic scRNA-seq reference

Data from *Okaty et al., 2020*, were downloaded from the NCBI GEO database (GSE144980). Raw sequencing reads were processed as above for the E14.5 scRNA-seq data using CellRanger. Cells were filtered using criteria as described in the original publication resulting in 2350 cells. The

'MapQuery' function in Seurat was used to project the adult scRNA-seq cells onto the E14.5 *Pet1* neuron scRNA-seq UMAP reference.

## Enhancer-gene (TAC-gene) association

scATAC-seq and scRNA-seq seurat objects were imported into SnapATAC (*Fang et al., 2021*). A gene expression profile for each cell based on the weighted average of the nearest neighboring cells (k = 15) was imputed based on the scRNA-seq dataset. From the integrated scATAC-seq and scRNA-seq datasets, a 'pseudo' multiomics cell is made that contains both gene expression and chromatin accessibility information. Then, logistic regression was performed to quantify the association between gene expression and the binarized accessibility state at putative enhancer elements in a 500 kb window centered at the TSS of each gene. The gene to putative enhancer associations were filtered with p-value ≤ 0.05.

## RNA-seq analysis

RNA-seq reads were aligned to UCSC mm10 using Hisat2 (v2.1.0) with a penalty of 12 for a non-canonical splice site, maximum and minimum penalty for mismatch of 1,0, and maximum and minimum penalty for soft-clipping of 3,0 parameters (*Kim et al., 2019*). Gene expression quantification and differential expression were calculated using Cufflinks v2.2.2 with a fold change cutoff of 1.5-fold and FDR cutoff of 0.05 (*Roberts et al., 2011*; *Trapnell et al., 2010*; *Trapnell et al., 2012*).

## CUT&RUN data processing and analysis

CUT&RUN data were processed using the standard ENCODE ChIP-seq software pipeline from the ENCODE Consortium. Sequencing reads were mapped to the UCSC mm10 by Bowtie v2.3.4.3. PCR duplicates were removed using samtools v1.9. The unique mapped reads were used to call peaks comparing TF-bound chromatin with an IgG control with SEACR using both stringent and relaxed threshold settings (*Meers et al., 2019*). The stringent threshold uses the peak of the total signal curve and the relaxed threshold uses the total signal of the 'knee' to the peak of the total signal curve. Both threshold settings have been shown to have high sensitivity and specificity (*Meers et al., 2019*). BEDTools was used to find shared intersections between identified peaks (1 bp minimum overlap). Peak annotations were performed using ChIPseeker v1.28.3. CUT&RUN data for Pet1 and Lmx1b were normalized together for visualization using the ChIPseqSpikeInFree method to determine the scaling factor for adjusting signal coverage across the genome (*Jin et al., 2020*).

## Footprinting analysis

TF footprints were identified using RGT-HINT (*Gusmao et al., 2016*; *Li et al., 2019*). First, we searched for motifs from HOCOMOCO v11 within the set of differentially accessible ATAC-seq peaks for each TF knockout condition (*Kulakovskiy et al., 2018*). Then, differential footprints were called on uniquely mapped paired-end ATAC-seq reads with bias correction for the Tn5 transposase within the set of differentially accessible ATAC-seq peaks comparing control to each TF knockout condition.

## Genomic visualization

ATAC-seq, ChIPmentation, and CUT&RUN data were visualized in Integrative Genomic Viewer (v2.9.3). Bam files for all ATAC-seq and ChIPmentation datasets were normalized to 1× genome coverage using deeptools (v3.5.0) bamCoverage.

## Statistical analysis

No statistical methods were used to predetermine sample size. Measurements for all experiments were taken from distinct samples. No methods were used to determine whether the data conformed to the assumptions of the statistical method. Experimenters were blinded to genotype during immunofluorescence analysis of 5-HT pericellular baskets. No other blinding of experiments was performed. Clustering of scATAC-seq and scRNA-seq was performed in an unbiased manner. Cell subtypes were assigned after unbiased clustering. Nuclei that did not meet quality thresholds were removed from further analyses as described in the Results and Materials and methods.

## Data visualization

Data plots were generated using ggplot2 (v3.3.5) in R (v4.0.4) (*Wickham et al., 2016*) and Microsoft Excel. 5-HT neurotransmission diagram was created using Biorender.com.

## Code availability

No custom code was used in this study. Open-source algorithms were used as described in the analysis methods. Details on how these algorithms were used to analyze data are available from the corresponding author upon request.

## Acknowledgements

This research was supported by NIH grants R01 MH117643 and R01 MH062723 to ESD, and R01 MH125918 to ESD and WCS (co-l), F30 MH122173, and T32 GM007250 to XLZ, and the Uehara Memorial Foundation Fellowship (201940009) and Japan Society for the Promotion of Science Overseas Research Fellowship to NT, and T32GM008056 to MMK. We thank Katherine Lobur for assistance with mouse breeding and Laura Marsland for assisting with immunohistochemistry analysis of Tph2 in embryonic mouse hindbrain. We thank Dr Polyxeni Philippidou for helpful comments on the manuscript. We thank Dr Carmen Birchmeier for the anti-Lmx1b antibody. This work was supported by the Cytometry and Microscopy Shared Resource at Case Comprehensive Cancer Center, the CWRU Genomics Core, and the CWRU Applied Functional Genomics Core.

## Additional information

### Funding

| Funder | Grant reference number | Author |
|---|---|---|
| National Institute of Mental Health | R01 MH117643 | Evan S Deneris |
| National Institute of Mental Health | R01 MH062723 | Evan S Deneris |
| National Institute of Mental Health | F30 MH122173 | Xinrui L Zhang |
| National Institute of General Medical Sciences | T32 GM007250 | Xinrui L Zhang |
| Uehara Memorial Foundation | 201940009 | Nobuko Tabuchi |
| Japan Society for the Promotion of Science | Overseas Research Fellowship | Nobuko Tabuchi |
| National Institute of General Medical Sciences | T32 GM008056 | Meagan M Kitt |
| National Institute of Mental Health | R01 MH125918 | Evan S Deneris William C Spencer |

The funders had no role in study design, data collection and interpretation, or the decision to submit the work for publication.

### Author contributions

Xinrui L Zhang, Conceptualization, Formal analysis, Investigation, Methodology, Validation, Visualization, Writing - original draft, Writing - review and editing; William C Spencer, Conceptualization, Data curation, Formal analysis, Investigation, Methodology, Validation, Visualization, Writing - original draft, Writing - review and editing; Nobuko Tabuchi, Investigation, Methodology, Validation, Visualization, Writing - review and editing; Meagan M Kitt, Conceptualization, Writing - review and editing; Evan S Deneris, Conceptualization, Funding acquisition, Supervision, Writing - original draft, Writing - review and editing

## Author ORCIDs

Xinrui L Zhang http://orcid.org/0000-0002-7728-6970
William C Spencer http://orcid.org/0000-0002-9700-8011
Evan S Deneris http://orcid.org/0000-0003-4211-9934

## Ethics

All animal procedures used in this study were in strict accordance with the Guide for the Care and Use of Laboratory Animals of the National Institutes of Health. The protocol was approved by the Case Western Reserve University School of Medicine Institutional Animal Care and Use Committee (Animal Welfare Assurance Number A3145-01, protocol #: 2014-0044).

## Decision letter and Author response

Decision letter https://doi.org/10.7554/eLife.75970.sa1
Author response https://doi.org/10.7554/eLife.75970.sa2

## Additional files

### Supplementary files

• Transparent reporting form
• Source data 1. TAC-gene predictions.

### Data availability

Sequencing data have been deposited in GEO under accession code GSE185737.

The following dataset was generated:

| Author(s) | Year | Dataset title | Dataset URL | Database and Identifier |
|---|---|---|---|---|
| Zhang XL, Spencer WC, Tabuchi N, Deneris ES | 2022 | Reorganization of postmitotic neuronal chromatin accessibility for maturation of serotonergic identity | http://www.ncbi.nlm.nih.gov/geo/query/acc.cgi?acc=GSE185737 | NCBI Gene Expression Omnibus, GSE185737 |

The following previously published datasets were used:

| Author(s) | Year | Dataset title | Dataset URL | Database and Identifier |
|---|---|---|---|---|
| Donovan LJ, Spencer WC, Kitt MM, Eastman BA, Lobur KJ, Jiao K, Silver J, Deneris ES | 2019 | Lmx1b is required at multiple stages to build expansive serotonergic axon architectures | https://www.ncbi.nlm.nih.gov/geo/query/acc.cgi?acc=GSE130514 | NCBI Gene Expression Omnibus, GSE130514 |
| Wyler SC, Spencer WC, Green NH, Rood BD, Crawford L, Craige C, Gresch P, McMahon DG, Beck SG, Deneris ES | 2016 | Pet-1 Switches Transcriptional Targets Postnatally to Regulate Maturation of Serotonin Neuron Excitability | https://www.ncbi.nlm.nih.gov/geo/query/acc.cgi?acc=GSE74315 | NCBI Gene Expression Omnibus, GSE74315 |

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
