## [Editor Report]

This study will be of interest to developmental biologists who study the gene regulatory mechanisms necessary for induction and maintenance of post-mitotic neuronal identity. The authors generated a useful resource of genomic data and provided new insights into the dynamic regulation of accessible chromatin regions in post-mitotic serotonin (5-HT) neurons of the mouse hindbrain. This work proposes two transcription factors (Pet1, Lmx1b) as necessary for establishment and maintenance of accessible chromatin regions in serotonin neurons.

---

## [Decision Letter]

**Decision letter after peer review:**

Thank you for submitting your article "Reorganization of postmitotic neuronal chromatin accessibility for maturation of serotonergic identity" for consideration by *eLife*. Your article has been reviewed by 3 peer reviewers, including Paschalis Kratsios as the Reviewing Editor and Reviewer #1, and the evaluation has been overseen by Catherine Dulac as the Senior Editor.

This work mapped accessible chromatin regions and gene expression using bulk and single cell analysis, performed chromatin immunoprecipitation studies of Pet-1 and Lmx1b transcription factors and investigated effects of Pet1 and Lmx1b loss of function on chromatin accessibility. The study generated a useful resource of genomic data and provided new insights into the dynamic regulation of accessible chromatin regions in postmitotic neurons. Technically, this is a tour-de-force effort.

Essential revisions:

The reviewers think the manuscript does not require additional experimental data, but it does require additional analysis of the existing data (as listed in the detailed reviewer comments below) and major revisions of the text.

1. While the manuscript provides important new data, the model how serotonergic neuron specification is controlled by Pet1 and Lmx1b remains unclear. Most importantly, the authors need to integrate better the single cell analysis with the global analysis of chromatin accessibility and TF binding data. Specifically, it would be quite interesting to address whether the pan-serotonergic selector factors also contribute to subtype diversification.

2. Another important point that deserves more attention is the analysis of differentially enriched motifs in neuronal subtypes and along the temporal trajectory. We also agree that the discrepancy between proximal vs distal TF binding and changes in chromatin accessibility should be better analyzed and highlighted in the manuscript. On the other hand, we are not sure whether correlative analysis between Pet1/Lmx1b binding sites and gene expression will be that informative, as control of gene expression is typically a result of complex integrated regulatory inputs from multiple distal enhancers.

3. We would like to see a better description of the 11 single cell clusters, and it would be great if the authors attempted to correlate the cluster identity with published adult serotonergic single cell data. However, we do understand that such analysis might be challenging and hard to interpret, so we do not think that the publication should be made conditional on this analysis.

4. Despite the wealth of new genomic data, the focus of the manuscript is somewhat elusive. The goal seems to be to study the mechanisms that control chromatin accessibility in postmitotic serotonergic neurons during their subtype diversification. The manuscript will benefit from clearer writing, explicit stating what hypotheses are being tested and better explanation of the rationale for the performed experiments.

*Reviewer #1 (Recommendations for the authors):*

1. Can the "pioneer" activity of Pet1 (or Lmx1b) be separated from its transcriptional activity? For example, ATAC-Seq and CUT&RUN data show that open chromatin regions are bound by Pet1. So, if Pet1 binds at a single genomic location upstream of gene x, does this binding event triggers the chromatin to open AND transcription to start? And what happens in cases where multiple binding peaks for Pet1 are found upstream of a gene? Some peaks are responsible for opening chromatin, while others for transcriptional activation?

2. Addressing the following questions by editing the text and/or re-analyzing existing data (RNA-Seq, ATAC-Seq, and Cut&RUN) could help the reader: Do all known Pet1 targets genes from previous RNA-Seq in serotonin neurons also require Pet1 to "open" their cis-regulatory regions? Are there any Pet1 targets whose expression is affected by Pet1 inactivation, but Pet1 does not open their cis-reg region? Lastly, are there any genes for which Pet1 only exerts a "pioneer function"? That is, it only opens their chromatin, but does not control directly the expression of these genes?

The same questions apply to Lmx1b.

3. It is very intriguing that the expression of 5-HT genes (Tph2, Slc6a4, Slc22a3) is low or undetectable at e14.5 despite their "open" chromatin status at this stage. Another intriguing observation comes from a previous study by the authors reporting that Pet1 switches its targets postnatally (Wyler et al., J Neurosci. 2016). One of the genes that is initially responsive to Pet1 is Tph2, but then, at postnatal stages, Tph2 expression does not depend on Pet1. Do the current data (e.g. CUT&RUN, ATAC-Seq, ChIP-seq) on WT and Pet1 cKO neurons help explain how Pet1 switches its targets in 5-HT neurons? For example, is Pet1 required to open the chromatin surrounding Tph2 and also activate Tph2 transcription early, but at later postnatal stages Pet1 only opens the chromatin for other TFs to activate Tph2 transcription?

4. Are ATAC-seq experiments quantitative enough to allow for comparisons between genotypes? For example, the height of each peak is compared between WT and cKO conditions in Figures 5e and 6i,j.

5. The authors discovered a new function for Pet1 and Lmx1b and therefore propose that these TFs define a previously unrecognized category of TFs termed "postmitotic neuron chromatin reorganizers" (PnCRs) that act specifically during postmitotic neuron maturation to shape the accessible chromatin landscapes and thereby select enhancers required for acquisition of specialized neuronal identities. While it is tempting to "name" the TFs, I would suggest to simply "name/state" their new function, especially in light of the fact that Pet1 and Lmx1b have already been termed "terminal selectors". To avoid confusion and help the reader, the authors should consider something along the following lines: Pet1 and Lmx1b exert a dual function: they not only act as terminal selectors, but also as chromatin reorganizers in postmitotic 5-HT neurons.

*Reviewer #2 (Recommendations for the authors):*

The authors should discuss the identity of the 11 major subtypes with unique chromatin accessibility patterns (Figure 2d; Supplemental Figure 2b) identified by scATAC and RNAseq analysis. How many of the clusters can be mapped on known serotonergic subtypes? How well do the subtypes map on published scRNAseq serotonergic neuron data?

Figure 3c: numbers do not match – gained and lost TACs do not add up to the numbers of peaks detected.

Figure3e: It is unclear how the enhancer-gene association was performed for this data. Were enhancer-gene associations used from the sc-seq data? Those were performed at e14.5 and would not have some of the peaks at e11.5 and e17.5 used in this plot. Were new enhancer-gene associations created from the temporal data- in this case this plot is making a circular argument (if the enhancer-gene associations are made based on co-regulation between expression and accessibility).

Figure 3i: needs more clear labeling of each of the gene sets in the four heatmaps

Figure 4a: pink dots in PCA seem extra?

Figure 4a: the PCA analysis shows that TACs are dominantly influenced by neuronal maturation, but such analysis does not reveal whether the Pet1-/- neurons changed identity. Instead, the authors should perform PCA analysis at one timepoint and include Pet1 negative wt neurons in the PCA analysis. This will address whether Pet1-/- neurons are more similar to Pet1 positive or Pet1 negative wt neurons.

Figure 4c-e: The analysis and interpretation of the correlation of chromatin accessibility with Pet1 binding in wt and mutant cells is extremely confusing. It is not clear what the main conclusion of this section is and what data support the conclusions. Figure 4c shows all TACs at different ages and plots their change in accessibility – the overall decrease of accessibility in Pet1-/- neurons is striking and surprising. Is this perhaps a technical issue? It is hard to believe that the loss of Pet1 TF that binds only to a subset of accessible sites results in a global ~4 fold reduction in chromatin accessibility. What is the number of TACs that gain, lose or do not change in Pet1-/- neurons? Is it correct that only 200 TACs are Pet1 regulated (i.e. bind Pet1 and change in KO)? How many Pet1 binding sites there are? How many of these binding sites are in TACs and how many are outside TACs? How are "Pet1-dependent" TACs defined?

The authors conclude that Pet1 is involved both in opening chromatin and in maintaining open chromatin. What differentiates Pet1 binding sites that gain accessibility compared from the sites that have high accessibility already at e11 (before Pet1 binds) or from the sites that loose accessibility in the conditional Pet1 knockout? Is it possible that sites that exhibit lower accessibility in conditional mice had not acquired full accessibility at e12.5 (i.e. the data cannot be interpreted as evidence for the role of TFs in the maintenance of TACs)

Figure 5: Pet1 cKO has a stronger effect on chromatin accessibility than the Pet1 full knockout. Is this a technical issue? If not, what are the chromatin regions affected in the cKO but not in full KO- are these enhancers based on chromatin profile (generated in Figure 1 at e14.5)? How do you explain these results?

The authors should integrate better the single cell analysis with analysis of Pet1 and Lmx1b binding and mutants. Do Pet1 and Lmx1b regulate TACs and expression levels of subtype specific genes in addition to pan-Pet1 lineage genes? Are these motifs co-enriched with other, more cell type specific motifs at subtype-specific peaks?

While majority of Lmx1b co-binds with Pet1, majority of Pet1 binds independently from Lmx1b. The data on Lmx1b and Pet1 binding overlap should be presented as a scatter plot rather than Venn diagrams. Are co-bound TACs regulated in a different way compared to TACs bound by Pet1 or Lmx1b only?

Considering that Pet1 expression depends on Lmx1b, the authors should address whether Pet1 expression is lost in Lmx1b CKO or whether it is maintained independently of Lmx1b.

Figure 7e: Legend says "Differential expression (from E11.5 to E17.5) of the genes associated with Pet1- occupied TACs in Pet1-cKO vs. wildtype 5-HT neurons (left)…", however the y-axis in plot is labeled "e17.5 KO/wt". If the plot is correctly labeled in the figure, it seems that genes associated with TACs are not changing expression in expected patterns. However, it does look like some synaptic genes do show reduced expression (Figure 7 and supp Figure 7). It would be interesting to know how gene expression is affected globally in cKO and DKO conditions, and if the genes affected are mainly pan-pet1 lineage genes (like 5-HT, synaptic, etc) or also subtype specific genes.

*Reviewer #3 (Recommendations for the authors):*

1. In general, I felt the work, as it is presently written, does not flow very well, going back and forward in stages and techniques. I am aware it is a lot of data and different methods but maybe it would be better organized as: (1) longitudinal data e11, e14, e17; (2) mutants longitudinal 3) e14.5 single cell data

2. I find authors often focus on specific examples from already well-characterized targets of Pet1 and Lmx1b, such as 5HT pathway genes but I miss a more unbiased analysis. Also, it would be nice to include a more detailed and integrated analysis of the embryonic scATACseq and scRNAseq with the adult scRNAseq (Okaty et al. and/or Ren et al. reports) as well as a more profound study of differential TF expression and TF motif enrichment in different TACs of different 5HT neuron subclasses, that could maybe help identify new TFs involved in subtype specification. Similarly, those analysis could also be done comparing differential TF expression and motif enrichment in differentially accessible TACs along time (e11, e14 and e17 data) to try to unravel TFs involved in maturation or in specific transient events.

3. The abstract mentions several conclusions, I find some of them are not really novel or lack enough data to support them:

a) "heterogeneous chromatin landscapes are established early": this has been already shown for other neuronal types.

b) "reveal the regulatory programs driving subtype identities of Pet1-lineage

neurons that generate serotonin (5-HT) neurons": I don´t think present data identify regulatory programs (that is networks of TFs and their targets).

c) Dynamic enhancers in postmitotic neurons have been already shown.

d) "We find that Pet1 and Lmx1b control chromatin accessibility to select Pet1-lineage specific enhancers for 5-HT neurotransmission and synaptogenesis" The description of changes in chromatin accessibility is interesting, however, the role of Pet1 and Lmx1b controlling gene expression of 5HT neurotransmission and synaptogenesis is not new.

e) "Required to maintain accessible chromatin in maturing neurons", also interesting however both TFs are already described to be required for gene expression maintenance

4. Regarding the mouse model: what is the % of 5HT neurons that are Pet1:YFP negative in each analysed time point? and vice versa, are all Pet1:YFP cells 5HT positive?

5. Why scATAC approach identifies 60,117 accessible sites not identified in the bulk Pet1? Are those uniquely identified TACs present only in subpopulations of Pet1 positive cells?

6. How do the 11 chromatin clusters of Pet1 scATAC correlate with adult scRNAseq data? It would be nice to have an integrated analysis of the embryonic and adult scRNAseq and the scATACseq data (including Okaty et al. and/or Ren et al. reports) to find the embryonic chromatin correlates of the adult 5HT subtypes.

7. Motif enrichment in scATAC clusters (Sup figure 3c). (1) Can the authors speculate on why ETS motif is only enriched in Tph2 cluster and LIM motif in Lrfn5 cluster? (2) could this motif enrichment analysis be combined with cluster specific or enriched TF expression for each family to identify TF candidates for subtype specification? For example, authors mention enriched expression of nkx6.1 for one cluster, which also shows enrichment in NK motifs (this double match TF expression with TF motif is very interesting and should be more clearly mentioned). Similar correlation analysis should be done for other TFs, TF motifs and clusters.

8. Sup figure 4j: motif enrichment of gained and lost TACs. Are the same motifs present in gained from e11 to e14 than those gained from e14 to e17? and the same question applies for the lost motifs. And as mentioned already, it would be advisable to merge TF expression dynamics with TF motif enrichment dynamics.

9. How does gene expression, gains and losses between wt and Pet1 mutants at different stages correlate with TACs: (1) What is the % of genes downregulated in Pet1 mutant compared to control that shows at least one assigned TAC in wt scATACseq or in bulk atacseq (2) from those, what is the % of downregulated genes in Pet1 mutant with TAC that lose accessibility in at least one assigned TAC? Similar analysis will be interesting for the >1000 genes with gained expression in e11.5 Pet1 mutants reported in Wyler et al. 2016.

10. Are genes associated with TACs lost in Pet1 mutant expressed in specific clusters of 5HT or broadly expressed in 5ht neurons? are they 5HT specific?

11. Could authors speculate why Pet1-/- phenotype is milder than Pet1cKO? (372 TACs less accessible and 3 more accessible compared to 1834 and 391 in cKO), is there any compensatory effect in the regular mutant? How do these 372 TACs and 3 TACs and their respective assigned genes match with DEX genes in Pet1 mutants at e14?

12. What genes are assigned to the 391 TACs gained in Pet1cKO? do they correlate with gained gene expression in this mutant background? what motifs are enriched in these TACs? what GO terms?

13. Most differential TACs in Pet1 lineage are found in distal intergenic or introns, and show enhancer chromatin marks, however CUT and RUN binding for both Pet1 and Lmx1b is found preferentially at proximal regions (promoters). Could authors specify de % of proximal TACs with Pet1 or Lmx1b binding compared to % of distal TACs with binding? Is motif enrichment in TACs different in proximal and distal TACs? What is the mechanistic explanation of this discrepancy between CUT and RUN binding and differential TACs and lost TACs in the mutants?

14. Donovan et al. showed Lmx1b is upstream Pet1 expression, at least in some 5HT neurons, Is the effect on chromatin accessibility of Lmx1b mutant mediated, at least in part, by the loss of Pet1? How many lost TACs in Lmx1b are bound by Pet1 but not by Lmx1b?

15. I would remove the last section of the manuscript, as I don´t think the data support a role in synaptogenesis: Authors mention vglut3 and syt1 as panneuronal genes, however they are neuronal effector genes not expressed in all neurons but in specific neurons subtypes. Moreover, they are not involved in synaptogenesis but in synaptic function and neurotransmission. Finally, the analysis of pericellular baskets does not make much sense as these mutants show defects in axon projections to start with.

---

## [Author Response]

Essential revisions:The reviewers think the manuscript does not require additional experimental data, but it does require additional analysis of the existing data (as listed in the detailed reviewer comments below) and major revisions of the text.1. While the manuscript provides important new data, the model how serotonergic neuron specification is controlled by Pet1 and Lmx1b remains unclear. Most importantly, the authors need to integrate better the single cell analysis with the global analysis of chromatin accessibility and TF binding data. Specifically, it would be quite interesting to address whether the pan-serotonergic selector factors also contribute to subtype diversification.

In response to this suggestion, we have added into the manuscript figure panels to describe the percentage of Pet1- and/or Lmx1b-regulated TACs that are subtype specific vs. common to multiple subtypes. These results show that Pet1 and Lmx1b both regulate some subtype-specific TACs and common TACs. We agree that it would be very interesting to test whether either TF contributes to subtype diversification. However, the focus of this work is on TF regulation of chromatin accessibility in postmitotic neurons, and thoroughly investigating TF regulation of subtype diversification is beyond the scope of this manuscript.

2. Another important point that deserves more attention is the analysis of differentially enriched motifs in neuronal subtypes and along the temporal trajectory. We also agree that the discrepancy between proximal vs distal TF binding and changes in chromatin accessibility should be better analyzed and highlighted in the manuscript. On the other hand, we are not sure whether correlative analysis between Pet1/Lmx1b binding sites and gene expression will be that informative, as control of gene expression is typically a result of complex integrated regulatory inputs from multiple distal enhancers.

We agree that these phenomena are very intriguing while also sharing the view that we are not sure whether correlation between TF binding sites and gene expression is all that informative. We have made an effort to give this more attention. Please see our detailed responses below in our responses to Reviewer #3 major points 8 and 13.

3. We would like to see a better description of the 11 single cell clusters, and it would be great if the authors attempted to correlate the cluster identity with published adult serotonergic single cell data. However, we do understand that such analysis might be challenging and hard to interpret, so we do not think that the publication should be made conditional on this analysis.

This is a good suggestion as it would be interesting to relate our developmental subtype results to published adult subtypes. As the reviewers note, this is a difficult analysis due in particular to the significant time gap between the datasets. To incorporate this suggestion, we focused on the adult serotonergic scRNA-seq data from Okaty et al. (2020) since they profiled the highest number of cells. Cell number is critical to identify any patterns that could emerge from projecting data from one dataset to another. Please see specific responses to reviewer #2 comment 1 below for further details on our approach and results.

4. Despite the wealth of new genomic data, the focus of the manuscript is somewhat elusive. The goal seems to be to study the mechanisms that control chromatin accessibility in postmitotic serotonergic neurons during their subtype diversification. The manuscript will benefit from clearer writing, explicit stating what hypotheses are being tested and better explanation of the rationale for the performed experiments.

The goal of the manuscript is to study the mechanisms that control chromatin accessibility in serotonergic neurons during their early postmitotic maturation, which is a period predominated by axon and synapse formation. We believe the focus became confusing because we introduced some single cell datasets early in the manuscript. The main purpose of the single cell analyses were to (1) demonstrate that chromatin accessibility is significantly heterogeneous not only among different neuron-types in the brain but also within Pet1 lineage neurons that will mature to become 5-HT neuron subtypes, (2) show that there is a strong correlation between chromatin accessibility and gene expression, and (3) associate specific TACs to specific genes that can then be subsequently be used during bulk ATAC-seq and RNA-seq analyses to understand the temporal maturation of chromatin.

In response to the reviewers’ feedback we clarified our writing in the summary to better explain the rationale for the experiments we performed.

Reviewer #1 (Recommendations for the authors):1. Can the "pioneer" activity of Pet1 (or Lmx1b) be separated from its transcriptional activity? For example, ATAC-Seq and CUT&RUN data show that open chromatin regions are bound by Pet1. So, if Pet1 binds at a single genomic location upstream of gene x, does this binding event triggers the chromatin to open AND transcription to start? And what happens in cases where multiple binding peaks for Pet1 are found upstream of a gene? Some peaks are responsible for opening chromatin, while others for transcriptional activation?

We believe the simplest interpretation of our findings is that Pet1 is able to open partially or completely inaccessible chromatin and subsequently engage in transcriptional activation. However, the details of Pet1 action are likely more complex and will require future study to elucidate.

It is difficult to definitively separate the chromatin “pioneering” and transcriptional activities of TFs, but we do observe instances suggesting the two processes may be decoupled. For example, there is significant Pet1 and Lmx1b occupancy at gene promoters that correlates strongly with Pet1/Lmx1b dependent gene expression, yet ablating Pet1 and/or Lmx1b did not result in a significant loss of chromatin accessibility at those promoters, suggesting that factors other than Pet1 and Lmx1b could be responsible for maintaining accessibility in those TSS proximal regions. As another example, at E14.5 Pet1 and Lmx1b are detected in all *Pet1* neurons by scRNA-seq and the TACs of 5-HT neurotransmission genes *Tph2*, *Slc6a4*, and *Slc22a3* are in an accessible state, yet the expression of these genes is low or undetected in many individual *Pet1* neurons, suggesting that there is a delay in Pet1/Lmx1b transactivation and chromatin opening preceded gene transcription.

The reviewer raises a number of other interesting questions, some of which cannot be answered definitively with our present findings. For example, we do not believe our findings shed light on whether Pet1 occupancy at some of an array of binding sites strictly controls opening while others strictly control transcriptional activation. It seems addressing this question would require systematic editing of individual binding sites, in vivo, followed by chromatin mapping and expression analyses.

Single cell studies of dividing cells (Ma et al., 2020; Trevino et al., 2021) drew a similar conclusion that chromatin accessibility often occurs prior to mRNA expression.

The reviewer’s proposal that Pet1 binding at different sites could have different functions (some for opening chromatin, while others for transcriptional activation) is a very intriguing idea. However, it is difficult to test this idea because TAC regions usually contain the binding sites of multiple TFs, so ablating the TF or mutating a binding site of the TF that is maintaining chromatin access also prevents the regional binding of other TFs that could be responsible for gene transcription.

2. Addressing the following questions by editing the text and/or re-analyzing existing data (RNA-Seq, ATAC-Seq, and Cut&RUN) could help the reader: Do all known Pet1 targets genes from previous RNA-Seq in serotonin neurons also require Pet1 to "open" their cis-regulatory regions? Are there any Pet1 targets whose expression is affected by Pet1 inactivation, but Pet1 does not open their cis-reg region? Lastly, are there any genes for which Pet1 only exerts a "pioneer function"? That is, it only opens their chromatin, but does not control directly the expression of these genes?The same questions apply to Lmx1b.

Our present findings suggest that not all known Pet1 target genes in serotonin neurons require Pet1 to open their cis-regulatory regions. There are Pet1 targets whose expression is affected by Pet1 inactivation, but Pet1 does not appear to open their associated CREs. Conversely, there are also examples for which Pet1 only exerts a “pioneer function” and does not control gene expression.

There are several possible explanations for these phenomena. First, not all differentially regulated genes in the Pet1 KO 5-HT neurons have direct nearby Pet1 occupancy, which suggests that some differential gene expression is due to either indirect or compensatory effects through the actions of other TFs, and therefore we do not expect to find Pet1 mediated chromatin accessibility changes near those genes. Secondly, Pet1 likely occupies CREs in combination with many other TFs similar to what we have shown between Pet1 and Lmx1b. In some instances, the removal of Pet1 is sufficient to destabilize the protein complex at a genomic region and result in the collapse of chromatin accessibility, whereas in other instances, chromatin accessibility in a genomic region is maintained in the absence of Pet1. We hypothesize that this difference can be attributed to the activities of the other regulatory factors that occupy CREs in combination with Pet1.

Lastly, as the reviewer proposed, it is very possible that the “pioneering” and transcriptional activities of TFs can be decoupled, though our data lack the resolution to definitively conclude this with certainty. To prioritize accuracy, we limited our TAC-to-gene association parameter so that TACs were only assigned to genes that were within 500kb, whereas enhancers are known to act across greater distances than 500kb, so we do not claim to have captured all TAC-to-gene interactions. In addition, CRE association with genes can be very fluid. For example, studies have shown that deletion of one TSS proximal enhancer can lead to compensatory activities from a more distal enhancer (e.g. Hong et al., 2008; Frankel et al., 2010; Perry et al., 2010). Therefore, it is possible that TACs dependent on Pet1 are not paired to all actual target genes.

Same for Lmx1b.

3. It is very intriguing that the expression of 5-HT genes (Tph2, Slc6a4, Slc22a3) is low or undetectable at e14.5 despite their "open" chromatin status at this stage. Another intriguing observation comes from a previous study by the authors reporting that Pet1 switches its targets postnatally (Wyler et al., J Neurosci. 2016). One of the genes that is initially responsive to Pet1 is Tph2, but then, at postnatal stages, Tph2 expression does not depend on Pet1. Do the current data (e.g. CUT&RUN, ATAC-Seq, ChIP-seq) on WT and Pet1 cKO neurons help explain how Pet1 switches its targets in 5-HT neurons? For example, is Pet1 required to open the chromatin surrounding Tph2 and also activate Tph2 transcription early, but at later postnatal stages Pet1 only opens the chromatin for other TFs to activate Tph2 transcription?

We agree that the dynamic changes in gene target sensitivity to Pet1 are interesting; this observation is what initially motivated our interest in probing the dynamics of postmitotic chromatin accessibility. Unfortunately, we do not believe our current findings offer any insight into the switching mechanism. We envision many potential mechanisms: (1) At around birth, Pet1 is no longer needed to remodel chromatin around some serotonergic genes such as Tph2 and other TFs such Lmx1b are sufficient for this job. To begin to investigate this idea we would need to target Pet1 at around P0 to see if loss of Pet1 at that time no longer had an effect on chromatin.

4. Are ATAC-seq experiments quantitative enough to allow for comparisons between genotypes? For example, the height of each peak is compared between WT and cKO conditions in Figures 5e and 6i,j.

ATAC-seq experiments are quantitative enough to allow for comparisons between genotypes. The height of each peak reflects the number of sequencing reads that map to that genomic region. Datasets are normalized to adjust for signal coverage across the genome. The height of visualized peaks can be compared between WT and cKO conditions.

5. The authors discovered a new function for Pet1 and Lmx1b and therefore propose that these TFs define a previously unrecognized category of TFs termed "postmitotic neuron chromatin reorganizers" (PnCRs) that act specifically during postmitotic neuron maturation to shape the accessible chromatin landscapes and thereby select enhancers required for acquisition of specialized neuronal identities. While it is tempting to "name" the TFs, I would suggest to simply "name/state" their new function, especially in light of the fact that Pet1 and Lmx1b have already been termed "terminal selectors". To avoid confusion and help the reader, the authors should consider something along the following lines: Pet1 and Lmx1b exert a dual function: they not only act as terminal selectors, but also as chromatin reorganizers in postmitotic 5-HT neurons.

We do appreciate the reviewer’s acknowledgment that we have discovered a chromatin reorganizing function for terminal selectors in maturing postmitotic neurons. We agree that our proposal to designate these as PnCRs, may be confusing since they already go by the descriptor, terminal selector. Yet, we maintain our findings reveal reorganization of postmitotic chromatin accessibility, which is a previously unrecognized functional attribute of how terminal selectors work. Thus, our findings advance the understanding of how terminal selectors determine specialized neuronal identities: terminal selectors, at least Pet1 and Lmx1b in mouse 5-HT neurons, remodel postmitotic accessible chromatin for selection of enhancers. Subsequently they transactivate via those selected enhancers, genes whose expression contributes to specialized neuronal identities. In this interpretation, Pet1 and Lmx1b do not possess dual and distinct terminal selector and chromatin reorganizing functions but rather the chromatin reorganizing function of these TFs is an inherent feature of their terminal selector action that operates prior to or together with transactivation. We believe it is reasonable and appropriate to discuss this duality of terminal selector action that has been inferred from the study of Pet1 and Lmx1b. How do these TFs reorganize postmitotic chromatin? One possibility is that they directly alter nucleosome positioning upon binding to ETS/HD motifs. Alternatively, we hypothesize that through their initial binding at ETS/HD motifs they recruit chromatin remodeling complexes that further open chromatin at these motifs for optimal sequence specific gene transactivation.

Reviewer #2 (Recommendations for the authors):The authors should discuss the identity of the 11 major subtypes with unique chromatin accessibility patterns (Figure 2d; Supplemental Figure 2b) identified by scATAC and RNAseq analysis. How many of the clusters can be mapped on known serotonergic subtypes? How well do the subtypes map on published scRNAseq serotonergic neuron data?

Okaty et al. performed 10X scRNA-seq at 1.5 to 2.5 mo on only DRN cells. The Ren et al. (2019) study used *Sert* positive neurons in adult dorsal and median raphe. Our analysis included cells from all raphe nuclei at E14.5. Since there is a large time gap between our datasets, it can be difficult for the integration algorithm to identify anchors between the two datasets. Nevertheless, we performed a reciprocal unimodal UMAP projection of our E14.5 scRNA with the Okaty et al. ~2mo scRNA data. First, we projected our scRNA clustering and UMAP structure onto the adult scRNA data. 93% of Okaty 10X DRN cells map to our Tph2/Slc6a4 (1099) and Pax5/En1 (1082) clusters. Since Tph2, Slc6a4, and the rhombomere 1 markers (e.g. En1) are highest in these two clusters, it is not surprising that the adult DRN neurons, which are only derived from isthmus rhombomere 1, are the most highly correlated with these clusters. Next, we projected the Okaty DRN scRNA clustering/UMAP onto our E14.5 scRNA. Since we did not have the exact clustering from Okaty et al., we analyzed their data according to their methods as precisely as possible, although it was not possible to reproduce the clustering exactly. Despite this issue, our clustering (14 clusters) and UMAP are strikingly similar to the results in Okaty et al. Then, we used the top cluster marker genes from Okaty et al., to determine which of our clusters are most similar to the Okaty clusters. By projecting the adult UMAP structure onto our E14.5 scRNA, most of our cells mapped to adult cluster 13, which are high Slc17a8 and low Tph2+ cells. This is reasonable since most of our cells have high Slc17a8 expression. The next largest number of cells mapped to adult cluster 1, which are high Tph2 expressing cells. 138 cells were mapped to cluster 7, which co-express Tph2 and Slc17a8. Lastly, 35 cells map to cluster 12, which are the Met expressing cells. For clarity, we added the projection of the Okaty data onto the embryonic clusters the manuscript in Figure 2 —figure supplement 1, since we do not have exactly the same clustering as published for the adult data.

Figure 3c: numbers do not match – gained and lost TACs do not add up to the numbers of peaks detected.

In our original figure panel, the gain/loss TAC numbers referred to the TACs with greater or equal than 2 fold change (increase = gain; decrease = loss) in accessibility and FDR <0.01, not the number of TACs that entirely appear/disappear between time points, so it did not add up to the total number of peaks called at each developmental time point. We have modified the figure to include both sets of numbers and included an explanation in the figure legends.

Figure3e: It is unclear how the enhancer-gene association was performed for this data. Were enhancer-gene associations used from the sc-seq data? Those were performed at e14.5 and would not have some of the peaks at e11.5 and e17.5 used in this plot. Were new enhancer-gene associations created from the temporal data- in this case this plot is making a circular argument (if the enhancer-gene associations are made based on co-regulation between expression and accessibility).

The enhancer-gene association was calculated using the E14.5 scRNA-seq and scATAC-seq datasets and tested on a separate set of bulk time course RNA-seq and ATAC-seq datasets to avoid making a circular argument. The single cell SnapATAC algorithm only paired a TAC with one or more genes if there was correlation between gene expression and accessibility within a single cell, so TACs with no differential accessibility between subtypes were not paired with a gene. Furthermore, to ensure the best accuracy we used the stringent parameter that a pairing is made only if the detected TAC was within 500kb of the gene. As a result, we only paired a small fraction of TACs with candidate genes. We then asked whether the accessibility of this specific subset of TACs is temporally correlated with gene expression in E11.5-E17.5 bulk datasets generated at another time and found a strong correlation.

Figure 3i: needs more clear labeling of each of the gene sets in the four heatmaps

We edited the figure to label the gene sets next to the heatmaps.

Figure 4a: pink dots in PCA seem extra?

They were indeed extra because of misalignment of the colored dots during figure compilation and we have corrected the figure.

Figure 4a: the PCA analysis shows that TACs are dominantly influenced by neuronal maturation, but such analysis does not reveal whether the Pet1-/- neurons changed identity. Instead, the authors should perform PCA analysis at one timepoint and include Pet1 negative wt neurons in the PCA analysis. This will address whether Pet1-/- neurons are more similar to Pet1 positive or Pet1 negative wt neurons.

We added PCA analysis at E14.5 with Pet1^neg^ neurons, wildtype *Pet1* neurons, and Pet1-/- *Pet1* neurons to Figure 4 —figure supplement 1.

Figure 4c-e: The analysis and interpretation of the correlation of chromatin accessibility with Pet1 binding in wt and mutant cells is extremely confusing. It is not clear what the main conclusion of this section is and what data support the conclusions. Figure 4c shows all TACs at different ages and plots their change in accessibility – the overall decrease of accessibility in Pet1-/- neurons is striking and surprising. Is this perhaps a technical issue? It is hard to believe that the loss of Pet1 TF that binds only to a subset of accessible sites results in a global ~4 fold reduction in chromatin accessibility. What is the number of TACs that gain, lose or do not change in Pet1-/- neurons? Is it correct that only 200 TACs are Pet1 regulated (i.e. bind Pet1 and change in KO)? How many Pet1 binding sites there are? How many of these binding sites are in TACs and how many are outside TACs? How are "Pet1-dependent" TACs defined?

We thank the reviewer for catching an error in Figure 4c and its legend that we presume has led to confusion. Figure 4c box plots display the distribution of the log2 fold change in accessibility of TACs showing differential accessibility between Pet1-/- vs. wildtype Pet1 neurons (not all TACs that were identified) at the three embryonic time points. We present the findings this way to make the point that there is an overall decrease in accessibility of TACs in Pet1-/- neurons compared to wildtype. This supports our interesting and novel conclusion that Pet1 is mainly required for opening or maintaining chromatin accessibility in postmitotic neurons. 372 TACs lost accessibility and 3 TACs gained accessibility (fold change ≥2, FDR<0.05) at E14.5 while the remaining E14.5 Pet1 neuron TACs did not change accessibility in a statistically significant manner. The reviewer is correct that only 200 of these TACs are Pet1 regulated (i.e. bind Pet1 and change in Pet1-/-). As shown in Figure 6 —figure supplement 1, Pet1 CUT&RUN peak calling identified 23,594 Pet1 binding sites, 22,285 of which are in TACs and 3280 are outside of TACs. We defined “Pet1-dependent” and “Pet1-regulated” TACs as TACs that show differences in accessibility (gain or loss) in the absence of Pet1.

The authors conclude that Pet1 is involved both in opening chromatin and in maintaining open chromatin. What differentiates Pet1 binding sites that gain accessibility compared from the sites that have high accessibility already at e11 (before Pet1 binds) or from the sites that loose accessibility in the conditional Pet1 knockout? Is it possible that sites that exhibit lower accessibility in conditional mice had not acquired full accessibility at e12.5 (i.e. the data cannot be interpreted as evidence for the role of TFs in the maintenance of TACs)

In this study we define maintenance as ongoing or continued requirement for Pet1 in sustaining wildtype chromatin landscape, in contrast to pioneer factors that induce “epigenetic memory” without ongoing expression of those factors. Therefore, if accessibility is lost in the cKO, we interpret that to mean that ongoing Pet1 activity is needed to maintain/sustain/preserve accessibility at the normal level.

About half of Pet1 or Lmx1b regulated TACs do not significantly change in accessibility between E11.5 and E14.5 in wildtype Pet1 neurons but lose accessibility in the cKOs, including the majority of 5-HT neurotransmission gene TACs. We added these numbers to the volcano graphs in Figures5 and 6.

Figure 5: Pet1 cKO has a stronger effect on chromatin accessibility than the Pet1 full knockout. Is this a technical issue? If not, what are the chromatin regions affected in the cKO but not in full KO- are these enhancers based on chromatin profile (generated in Figure 1 at e14.5)? How do you explain these results?

Our best hypothesis is that there is a compensatory effect to losing Pet1 after it has already been expressed and has activated other genes, which include other transcription factors. Only ~20% of Pet1 maintained TACs are directly occupied by Pet1 in wildtype neurons at E14.5 (and very few of the TACs that gain accessibility in Pet1-cKO have direct Pet1 occupancy), whereas 53% of Pet1-dependent TACs in KO are directly occupied by Pet1, which suggests that the larger number of TAC changes in the cKO are attributable to indirect effects.

Similar to ATAC-seq, we have observed in our lab differences between Pet1-cKO and Pet1-/- 5-HT neuron transcriptomic changes in RNA-seq experiments, which suggest that the difference between the two knockout models is unlikely to be a simple technical issue. We have also shown previously by knocking down Pet1 at different developmental time points that Pet1 switches transcriptional targets during different stages of 5-HT neuron development (Wyler et al., 2016), so it is possible that Pet1 regulates more TACs at later time points.

The chromatin regions affected in the cKO but not in germ line KO are predominantly (59%) active and poised enhancers distal to the TSS.

The authors should integrate better the single cell analysis with analysis of Pet1 and Lmx1b binding and mutants. Do Pet1 and Lmx1b regulate TACs and expression levels of subtype specific genes in addition to pan-Pet1 lineage genes? Are these motifs co-enriched with other, more cell type specific motifs at subtype-specific peaks?

The majority of the Pet1 or Lmx1b regulated TACs belong to pan-Pet1 lineage genes shared among multiple subtypes, but there are also subtype-specific genes that are regulated by the two TFs. 21% of Pet1 regulated TACs and 15% of the Lmx1b regulated TACs are subtype-specific.

We show motif activity enrichment for each scATAC cluster in Figure 2—figure supplement 2. We do not expect Pet1 (ETS) and Lmx1b (LIM-HD) motifs to be enriched in every cluster since they are equally expressed in all clusters.

While majority of Lmx1b co-binds with Pet1, majority of Pet1 binds independently from Lmx1b. The data on Lmx1b and Pet1 binding overlap should be presented as a scatter plot rather than Venn diagrams. Are co-bound TACs regulated in a different way compared to TACs bound by Pet1 or Lmx1b only?

We added to the manuscript a scatter plot showing the binding overlap between Pet1 and Lmx1b as suggested.

On average, in wildtype Pet1-lineage neurons co-bound TACs show greater ATAC-seq signal than TACs bound by only Pet1 or only Lmx1b. Loss of chromatin accessibility is seen in cKO conditions both for TACs that are co-bound and for those bound by only Pet1 or only Lmx1b.

**Author response image 1. sa2fig1:** 

Considering that Pet1 expression depends on Lmx1b, the authors should address whether Pet1 expression is lost in Lmx1b CKO or whether it is maintained independently of Lmx1b.

*Pet1* expression is reduced to 52% of wildtype level in the Lmx1b-cKO *Pet1* neurons based on our RNA-seq analyses at E17.5. However, the reduction in *Pet1* expression cannot entirely account for the dramatic changes in chromatin accessibility seen in the Lmx1b-cKO neurons because 1) significant residual Pet1 expression remains 2) Lmx1b-cKO neurons show 2823 differentially accessible TACs relative to wildtype neurons that were not observed in the Pet1-cKO (Figure 6 —figure supplement 1), including TACs associated with the 5-HT neurotransmission genes *Gchfr*, *Slc18a2*, and *Maoa*. We added a graph in Figure 6 —figure supplement 1 to show the expression of Pet1 in Lmx1b-cKO and the expression of Lmx1b in Pet1-cKO.

Figure 7e: Legend says "Differential expression (from E11.5 to E17.5) of the genes associated with Pet1- occupied TACs in Pet1-cKO vs. wildtype 5-HT neurons (left)…", however the y-axis in plot is labeled "e17.5 KO/wt". If the plot is correctly labeled in the figure, it seems that genes associated with TACs are not changing expression in expected patterns. However, it does look like some synaptic genes do show reduced expression (Figure 7 and supp Figure 7). It would be interesting to know how gene expression is affected globally in cKO and DKO conditions, and if the genes affected are mainly pan-pet1 lineage genes (like 5-HT, synaptic, etc) or also subtype specific genes.

The graphs show that the genes linked to TACs that lose accessibility in the Lmx1b-cKO (“Loss”) are downregulated in expression relative to genes linked to TACs that show no change in accessibility in the Lmx1b-cKO (“No Change”) (middle panel). Similarly, genes linked to TACs that lose accessibility in the DKO are downregulated in expression relative to genes linked to TACs with no change in accessibility in the DKO (right panel). These two associations were the only ones that were statistically significant and are consistent with the genes *Syt1* and *Slc17a8* that we used as examples.

Other correlations did not reach statistical significance, including Pet1-cKO Loss TACs and the expression of their associated genes. One possible explanation for this is that while the majority of Pet1- or Lmx1b-regulated TACs are likely enhancers, a significant percentage did not have common enhancer or promoter associated histone modifications (e.g. Figures 5d and 6e) and could therefore be silencers that negatively influence target gene expression, or insulators that do not directly influence expression. For example, we have previously found that the loss of Pet1 can lead to the activation of repressed genes (Wyler et al., 2016), suggesting that Pet1 could have repressor activities. Another possibility is that enhancer usage is dynamic in neurons. When one enhancer becomes decommissioned in the KO neuron, shadow enhancers that act redundantly on the same gene may compensate to maintain gene expression.

As suggested by the reviewer we added a panel to show how gene expression is affected globally in DKO condition in the figure supplement for Figure 7. The top biological function gene ontology terms related to the downregulated genes in DKO Pet1 neurons are indole-containing compound (e.g. serotonin) metabolic process and neuron projection guidance. The top biological function gene ontology terms related to the up-regulated genes are response to lipoprotein particle and membrane invagination. The most significant (lowest FDR) GO term is serotonin biosynthetic process.

Pet1-cKO and Lmx1b-cKO RNA-seq datasets were previously generated and analyzed in Donovan et al., 2019, so we did not include detailed analyses of these datasets in this paper. The genes affected in cKO and DKO are mostly genes shared among multiple subtypes of Pet1 lineage neurons but some Pet1 and Lmx1b regulated genes are subtype specific.

Reviewer #3 (Recommendations for the authors):1. In general, I felt the work, as it is presently written, does not flow very well, going back and forward in stages and techniques. I am aware it is a lot of data and different methods but maybe it would be better organized as: (1) longitudinal data e11, e14, e17; (2) mutants longitudinal 3) e14.5 single cell data

We have discussed this comment at length to consider the reviewer’s proposed reorganization. After careful consideration our view is that although there may be issues with flow in our chosen organization it appears to work best given the varied datasets and methods used. The reason that the single cell datasets were presented towards the beginning was that we used those data to link TACs to their putative regulated genes, and later used these TAC-to-gene pairings on bulk sequencing data to probe the importance of chromatin regulation for developmental/temporal gene expression trajectories. Single cell data gave us much higher resolution for enhancer-gene mapping than mapping using bulk ATAC-seq and RNA-seq data.

2. I find authors often focus on specific examples from already well-characterized targets of Pet1 and Lmx1b, such as 5HT pathway genes but I miss a more unbiased analysis. Also, it would be nice to include a more detailed and integrated analysis of the embryonic scATACseq and scRNAseq with the adult scRNAseq (Okaty et al. and/or Ren et al. reports) as well as a more profound study of differential TF expression and TF motif enrichment in different TACs of different 5HT neuron subclasses, that could maybe help identify new TFs involved in subtype specification. Similarly, those analysis could also be done comparing differential TF expression and motif enrichment in differentially accessible TACs along time (e11, e14 and e17 data) to try to unravel TFs involved in maturation or in specific transient events.

We maintain that Pet1 lineage neuron specific TACs are of interest because these determine overt serotonergic identity. As suggested by the reviewer, we added more detailed and integrated correlation of our embryonic scATAC-seq and scRNA-seq with the Okaty adult scRNA-seq (more discussion on this in response to reviewer 2 comment #1). We have included in Figure 2 —figure supplement 2 some data on differential TF expression and TF motif enrichment in TACs of different 5-HT neuron subclasses; while this is an interesting topic, the focus of this manuscript is on how Pet1 and Lmx1b regulate shared 5-HT neuron TACs that are important for 5-HT connectivity and neurotransmission.

We included in Figure 3 —figure supplement 1j the motifs that are differentially accessible along time to analyze the TFs involved in maturation. See response to comment #8 for more discussion on this.

3. The abstract mentions several conclusions, I find some of them are not really novel or lack enough data to support them:a) "heterogeneous chromatin landscapes are established early": this has been already shown for other neuronal types.

Heterogeneity in chromatin landscapes has not been shown for 5-HT/*Pet1* neurons. With the exception of the very recent publication from Allaway et al. (2021) and Closser et al. (2022), studies on brain neuron chromatin diversity relied on single cell analyses of dissect bulk brain dissections, largely focusing on the differences between different neuronal and non-neuronal cell types instead of profiling the interesting heterogeneity that exists even within a single neurotransmitter subtype or single transcriptional factor lineage subpopulation. Thus, we maintain our presentation of heterogeneous chromatin landscapes within the single population of Pet1 lineage neurons distinguishes our work.

b) "reveal the regulatory programs driving subtype identities of Pet1-lineageneurons that generate serotonin (5-HT) neurons": I don´t think present data identify regulatory programs (that is networks of TFs and their targets).

We maintain that our analyses do in fact shed light on the Pet1 neuron gene regulatory network. First, our study identifies over 15,000 putative cis-regulatory regions that uniquely define the *Pet1* neurons and are not found in other hindbrain neurons or annotated ENCODE ATAC-seq datasets from many tissues and cell types. We then used motif analysis to identify candidate TFs that drive Pet1 neuron identity. This analysis revealed that ETS motifs and LIM-homeodomain motifs, among many TF families, are highly enriched in *Pet1* neuron specific TACs. We then associated Pet1 neuron subtype specific cis-regulatory elements with subtype specific transcription factors and downstream target genes using motif discovery and network inference. We tested our enhancer-to-gene associations using independently generated bulk RNA-seq time course datasets and found a strong correlation between TAC accessibility and target gene expression. And lastly, as proof of principle we ablated two of these candidate TFs, Pet1 and Lmx1b, and show that they directly regulate neuronal chromatin accessibility. Lastly, we show that the expression of genes associated with Lmx1b and Pet1 maintained TACs are overall significantly downregulated in expression in the absence of Lmx1b and Pet1.

c) Dynamic enhancers in postmitotic neurons have been already shown.

We agree that dynamic enhancer accessibility in postmitotic neurons has been reported in postnatal cerebellar development (Frank et al. 2015), Sst and Vip neurons (Stroud et al., 2020), and in developing forebrain tissue (Nord et al. 2013). Although we realize we were not the first to report dynamic enhancers in postmitotic neurons, enhancers have not been mapped, or shown to be dynamic, for 5-HT or Pet1-lineage neurons. Our study reports dynamic enhancer accessibility in embryonic Pet1 lineage neurons, which we presented as a basis to predict (or raise the possibility) that certain factors exist to remodel postmitotic chromatin and thereby support postmitotic neuron maturation.

d) "We find that Pet1 and Lmx1b control chromatin accessibility to select Pet1-lineage specific enhancers for 5-HT neurotransmission and synaptogenesis" The description of changes in chromatin accessibility is interesting, however, the role of Pet1 and Lmx1b controlling gene expression of 5HT neurotransmission and synaptogenesis is not new.

We certainly agree that we have previously shown using RNAseq and RT-qPCR that Pet1 and Lmx1b control expression of 5-HT neurotransmission genes such as Tph2 and Slc6a4. So yes, we have already shown that these TFs control 5HT neurotransmission gene expression. We note however, that none of our previous papers have focused on gene expression underlying synaptogenesis or synapse functions. What is new in our present work is a deeper mechanistic understanding of how these TFs control gene expression: we show that Pet1 and Lmx1b remodel postmitotic chromatin accessibility to enable the expression of neurotransmission genes and genes needed for synaptogenesis in maturing 5-HT neurons. The identification of TFs that remodeling postmitotic neuronal chromatin is new and this discovery was made possible by our earlier published work at the level of 5HT neurotransmission gene expression.

e) "Required to maintain accessible chromatin in maturing neurons", also interesting however both TFs are already described to be required for gene expression maintenance

Yes, it is true that we have previously reported findings indicating that Pet1 and Lmx1b maintain the expression of 5-HT neurotransmission genes such as Tph2. That is not the point of our current paper. The current study builds on this knowledge to put forth a new and surprising finding: these TFs maintain gene expression by maintaining accessible CREs in postmitotic neurons. The remodeling of accessible chromatin by Pet1and Lmx1b is the most novel and interesting of our findings in our present manuscript and was made possible by our previous studies of these TFs.

Our findings suggest that by controlling chromatin accessibility, these TFs not only directly modulate transcription activation as previously shown, but also impact the global landscape of active CREs and available TF binding sites which, in turn, can alter the global network of regulatory factor inputs. It broadens our understanding of how neuronal identity is maintained in postmitotic neurons (e.g. not just maintenance of transcription activation but also maintenance of CRE landscape).

4. Regarding the mouse model: what is the % of 5HT neurons that are Pet1:YFP negative in each analysed time point? and vice versa, are all Pet1:YFP cells 5HT positive?

The number of 5-HT neurons that are Pet1YFP positive is nearly 100% at all three time point. Therefore the % that are Pet1YFP negative is close to zero or is zero. At 11.5, <5% of Pet1YFP+ cells are 5-HT+, at E14.5 roughly 22% of Pet1YFP+ cells are 5-HT+, and at E17.5, roughly 42% of Pet1YFP+ cells are 5-HT+. In adult 5-HT neurons nearly all Pet1YFP+ cells are 5-HT+. This is consistent with our scRNA-seq findings showing that not all Pet1-lineage neurons are Tph2 positive at embryonic time points.

Our findings are consistent with Alonso et al. (2013) and the Allen Brain Atlas: Tph2 expression is very sparse and limited to a few rostral neurons at E11.5 and then gradually expands throughout Pet1-lineage neurons at E14.5 and E17.5 and into adulthood.

5. Why scATAC approach identifies 60,117 accessible sites not identified in the bulk Pet1? Are those uniquely identified TACs present only in subpopulations of Pet1 positive cells?

Yes, 63% (38043/60117) of the scATAC peaks not identified in the bulk E14.5 WT ATAC are present only in subpopulations of Pet1+ cells. The remainder may not be called peaks in the bulk ATAC due to more stringent peak calling parameters compared to standard scATAC-seq methods.

6. How do the 11 chromatin clusters of Pet1 scATAC correlate with adult scRNAseq data? It would be nice to have an integrated analysis of the embryonic and adult scRNAseq and the scATACseq data (including Okaty et al. and/or Ren et al. reports) to find the embryonic chromatin correlates of the adult 5HT subtypes.

See response to Reviewer 2 first point.

7. Motif enrichment in scATAC clusters (Sup figure 3c). (1) Can the authors speculate on why ETS motif is only enriched in Tph2 cluster and LIM motif in Lrfn5 cluster? (2) could this motif enrichment analysis be combined with cluster specific or enriched TF expression for each family to identify TF candidates for subtype specification? For example, authors mention enriched expression of nkx6.1 for one cluster, which also shows enrichment in NK motifs (this double match TF expression with TF motif is very interesting and should be more clearly mentioned). Similar correlation analysis should be done for other TFs, TF motifs and clusters.

Bulk ATAC-seq results suggest that ETS motifs are highly enriched in general in *Pet1-*lineage neurons compared to other cell types. Within Pet1 neurons, the scATAC-seq suggests that one subpopulation (“Tph2/Slc6a4” cluster) has slightly more accessible (<2 fold) ETS motifs than the other clusters, but that does not mean that other clusters have no accessible ETS motifs. Similar scenario applies to the LIM homeodomain motifs in the Lrfn5/Nr2f2 cluster. The Tph2/Slc6a4 cluster maps to rostral Pet1 neurons with higher Tph2 and Slc6a4 expression. Rostral 5-HT neurons are known to mature and synthesize 5-HT a day earlier than caudal 5-HT neurons. Pet1::EYFP is also activated earlier in the rostral 5-HT neurons. We would therefore speculate that this cluster is capturing that subpopulation that is advancing the fastest in development, perhaps due to earlier Pet1 activity.

We present Figure 2 —figure supplement 2d as a list of TF candidates for 5-HT neuron subtype specification. In response to the reviewer’s suggestion we mentioned in the text a second TF, Meis2, from the single cell data in addition to Nkx6.1, which the reviewer found to be interesting. As the reviewer mentioned, there is a lot of data in this study, so for space saving reasons we chose to focus mostly on our bulk ATAC-seq analyses, as the TFs we tested, Pet1 and Lmx1b, are mainly responsible for the maturation of shared Pet1 neuron TACs. We show the candidate subtype specific TFs to emphasize that there is further heterogeneity even within Pet1-lineage neurons, which is an interesting topic for future studies.

8. Sup figure 4j: motif enrichment of gained and lost TACs. Are the same motifs present in gained from e11 to e14 than those gained from e14 to e17? and the same question applies for the lost motifs. And as mentioned already, it would be advisable to merge TF expression dynamics with TF motif enrichment dynamics.

Motifs present in gained from E11 to E14 are roughly similar to those gained from E14 to E17, the same for loss TACs. Bioinformatics correlation of motifs to gene expression changes is challenging because many TFs that are known to be active in developing 5-HT neurons belong to the same TF families. For example, Meis1, Meis2, Lmx1b, En1, En2, Pbx2, Hox factors, and POU domain factors are all homeodomain proteins that are expressed in Pet1 neurons. As another example, multiple Rfx-related factors are expressed in developing Pet1 neurons. However, these TFs with very similar binding motifs have different spatiotemporal expression patterns. For instance, Rfx2 and Rfx4 are downregulated from E11.5-E17.5, but Rfx1 is significantly upregulated, whereas Rfx7 and Rfx3 show little change in expression level. Because these TFs within the same or related families bind very similar motif sequences, it is difficult to assign specific genomic sites containing their cognate motif to the activity of a specific TF. Overall, we could detect a significant correlation between TF expression and motif accessibility if all members of the TF family show the same general trend in gene trajectories (e.g. correlation in downregulated *Sox2* expression and reduction in Sox family motifs within TACs from E11.5 to E17.5), but when multiple TFs have similar motifs, the results are difficult to interpret.

Additionally, it is possible for TF levels to not be tightly correlated with binding site accessibility level because some TFs may lack chromatin remodeling activity of their own such that its genomic binding is dependent on the priming of the chromatin or recruitment by other TFs and regulatory factors.

9. How does gene expression, gains and losses between wt and Pet1 mutants at different stages correlate with TACs: (1) What is the % of genes downregulated in Pet1 mutant compared to control that shows at least one assigned TAC in wt scATACseq or in bulk atacseq (2) from those, what is the % of downregulated genes in Pet1 mutant with TAC that lose accessibility in at least one assigned TAC? Similar analysis will be interesting for the >1000 genes with gained expression in e11.5 Pet1 mutants reported in Wyler et al. 2016.

65% (499) of Pet1 downregulated at E15.5 in PKO have at least one assigned TAC.

20% (98) of those genes have a TAC that loses accessibility in PKO/PCKO.

73% (734) of Pet1 upregulated at E15.5 in PKO have at least one assigned TAC.

5% (38) of those have a TAC that gains accessibility in PKO/PCKO.

10. Are genes associated with TACs lost in Pet1 mutant expressed in specific clusters of 5HT or broadly expressed in 5ht neurons? are they 5HT specific?

Of the TACs that are differentially accessible in the Pet1-cKO *Pet1* neurons, 1432 are shared/broadly expressed among Pet1 neurons and 391 are specific to one cluster. Of the TACs that are differentially accessible in the Pet1-/- *Pet1* neurons, 368 are shared and 20 are specific to one cluster.

Of the TACs that are differentially accessible in the Pet1-cKO *Pet1* neurons, 33% are “5-HT specific”. Of the TACs that are both differentially accessible in the Pet1-cKO *Pet1* neurons and contain Pet1 occupancy, 50% are “5-HT specific”. Of the TACs that are differentially accessible in the Pet1-/- *Pet1* neurons, 73% are 5-HT specific. Of the TACs that are both differentially accessible in the Pet1-/- *Pet1* neurons and contain Pet1 occupancy, 83% are 5-HT specific (Figure 4f).

11. Could authors speculate why Pet1-/- phenotype is milder than Pet1cKO? (372 TACs less accessible and 3 more accessible compared to 1834 and 391 in cKO), is there any compensatory effect in the regular mutant? How do these 372 TACs and 3 TACs and their respective assigned genes match with DEX genes in Pet1 mutants at e14?

Our hypothesis is described above in response to Rev 2 comment #9.

289 genes are linked to the 372 TACs that show loss in accessibility in Pet1-/- 5-HT neurons. Of these 289 genes, 35 are down-regulated in expression in Pet1-/- 5-HT neurons compared to wildtype neurons at E15.5. One of the 3 TACs that show gain in accessibility in Pet1-/- is a distal bivalent enhancer (based on ChromHMM) linked to the gene Rnf2. Rnf2 is down-regulated in expression in Pet1-/- 5-HT neurons at E15.5.

12. What genes are assigned to the 391 TACs gained in Pet1cKO? do they correlate with gained gene expression in this mutant background? what motifs are enriched in these TACs? what GO terms?

The 391 TACs gained in Pet1-cKO are mapped to 135 genes that are not significantly correlated with gained gene expression in the Pet1-cKO mutant background. These genes are most closely associated with the GO terms “negative regulation of cell volume”, “regulation of axon diameter”, “long-term synaptic potentiation”, and “potassium ion transport”, but none of these GO terms reached statistical significance. The 391 TACs are enriched for Rfx, Gata, HD, and ETS motifs.

13. Most differential TACs in Pet1 lineage are found in distal intergenic or introns, and show enhancer chromatin marks, however CUT and RUN binding for both Pet1 and Lmx1b is found preferentially at proximal regions (promoters). Could authors specify de % of proximal TACs with Pet1 or Lmx1b binding compared to % of distal TACs with binding? Is motif enrichment in TACs different in proximal and distal TACs? What is the mechanistic explanation of this discrepancy between CUT and RUN binding and differential TACs and lost TACs in the mutants?

Pet1 directly binds 59% of proximal promoter Pet1-dependent TACs and only 21% of the distal Pet1-dependent TACs. Lmx1b directly binds 32% of the proximal promoter Lmx1b-dependent TACs and only 9% of distal Lmx1b-dependent TACs.

Proximal and distal Pet1 and Lmx1b genome occupancy sites do show different nearby motif enrichments. Promoter proximal Pet1 occupied TACs are most highly enriched for ETS motifs and C2H2 zinc finger factor motifs, whereas distal Pet1 bound TACs are highly enriched for Rfx and homeodomain motifs. Similarly, promoter proximal Lmx1b bound TACs are enriched for C2H2 zinc finger factor motifs, CREB-related factor motifs, and ETS motifs, whereas distal Lmx1b bound TACs are also enriched for Rfx motifs and homeodomain motifs.

It’s been documented since the initial mapping of DNase-hypersensitivity sites that distal CREs are highly cell type selective compared to TSS-proximal CREs (e.g. Thurman et al., 2012), but the precise reason for this remains poorly understood. One possible mechanistic explanation is that promoter proximal TACs are occupied and more stably maintained by trans factors that sustain chromatin accessibility even in the absence of cell type specific pioneer TFs. We do not currently have enough evidence to draw more specific conclusions on what these factors are based on our data, as TF motif sites can be opened by chromatin remodeling factors other than the cognate TF itself.

14. Donovan et al. showed Lmx1b is upstream Pet1 expression, at least in some 5HT neurons, Is the effect on chromatin accessibility of Lmx1b mutant mediated, at least in part, by the loss of Pet1? How many lost TACs in Lmx1b are bound by Pet1 but not by Lmx1b?

*Pet1* expression is reduced to 52% of wildtype level in the Lmx1b-cKO *Pet1* neurons based on our RNA-seq analyses at E17.5. We added a panel into Figure 6 —figure supplement 1 to clarify this. However, the reduction in *Pet1* expression cannot completely account for the dramatic changes in chromatin accessibility seen in the Lmx1b-cKO neurons because 1) significant residual Pet1 expression remains 2) Lmx1b-cKO neurons show 2823 differentially accessible TACs relative to wildtype neurons that were not observed in the Pet1-cKO (Figure 6 – supplemental Figure 1), including TACs associated with the 5-HT neurotransmission genes *Gchfr*, *Slc18a2*, and *Maoa*.

Pet1 binds to 23% (699/3102) of LKO down TACs that are not bound by Lmx1b.

15. I would remove the last section of the manuscript, as I don´t think the data support a role in synaptogenesis: Authors mention vglut3 and syt1 as panneuronal genes, however they are neuronal effector genes not expressed in all neurons but in specific neurons subtypes. Moreover, they are not involved in synaptogenesis but in synaptic function and neurotransmission. Finally, the analysis of pericellular baskets does not make much sense as these mutants show defects in axon projections to start with.

Our reasoning is that for synaptogenesis to occur many effector genes must be activated to express presynaptic machinery e.g. Syt1. If this is prevented functional synapse formation is impaired. We showed that loss of Pet1 or Lmx1b impairs Syt1 chromatin accessibility and gene expression. Our interpretation is that these terminal selectors are needed to reorganize postmitotic chromatin to enable expression of genes needed for the formation of presynaptic machinery, hence synaptogenesis, but we agree with the reviewer that this wording choice can be confusing and have revised it to “synapse function” in the text. We would like to point out that while most axon projections to the dorsolateral septum fail to form some make it there. These successfully projections are the ones we investigated to determine whether they form pericellular baskets.